# SCAD: Super-Class-Aware Debiasing for Long-Tailed Semi-Supervised Learning

**Sunguk Jang**[*]
AITRICS

**Jinwoo Jeon**[*]
Korea University

**Byungjun Lee**
Korea University

## Abstract

In long-tailed semi-supervised learning (LTSSL), pseudo-labeling often creates a vicious cycle of bias amplification. Recent methods attempt to mitigate this issue via logit adjustment (LA). However, LA-based debiasing remains inherently hierarchy-agnostic and fails to account for semantic relationships between classes. We reveal a critical yet overlooked problem of *intra-super-class imbalance*, where semantically similar classes within a super-class are both highly confusable and locally imbalanced. This combination reinforces early mistakes, causing minority-class representations to be suppressed by their majority neighbors. To break this cycle, we propose Super-Class-Aware Debiasing (SCAD), a framework that performs dynamic, super-class-aware logit adjustment. SCAD leverages latent semantic structure to concentrate its corrective power on the most confusable groups, thereby resolving local imbalances. Extensive experiments demonstrate that SCAD achieves state-of-the-art performance. The code is available at https://github.com/aitrics-tom/SCAD.

## 1 Introduction

Deep learning models have achieved remarkable success on large, annotated datasets, but creating them is expensive (Deng et al., 2009; Lin et al., 2014). Semi-supervised learning (SSL) is a powerful approach to reduce this cost by leveraging a small labeled dataset alongside a large unlabeled dataset (Sohn et al., 2020; Berthelot et al., 2020). However, the performance of SSL is often hindered by the long-tailed distribution of real-world data (Figure 1a), where a few majority classes vastly outnumber many minority classes (Kang et al., 2020; Cui et al., 2019). This imbalance is particularly problematic because it creates a vicious cycle in long-tailed semi-supervised learning (LTSSL). This feedback loop then reinforces bias through biased pseudo-labels and causes severe performance drops for minority classes (Kim et al., 2020; Wei et al., 2021).

To break this vicious cycle, logit adjustment (LA) (Menon et al., 2021), a powerful method from supervised long-tail learning, has become the *de facto standard* for pseudo-label debiasing in LTSSL. LA applies a static, corrective offset to each class's logits, computed from the global class frequencies. The appeal of this approach lies in its theoretical guarantee. LA provides a *Fisher-consistent* correction for any given class prior with negligible overhead, making it an ideal foundational block for more complex scenarios. Building upon this foundation, recent state-of-the-art methods have made remarkable progress by deriving more sophisticated estimates of the true prior to leverage within the LA framework, such as by estimating the distribution mismatch (Wei & Gan, 2023) or measuring the classifier's intrinsic bias (Lee & Kim, 2024). However, while these methods innovate in how they estimate the global prior, the adjustment scheme itself remains the same as LA. It uses a single, uniform correction that is inherently blind to semantic relationships between classes.

In this paper, we uncover that this blindness to semantic relationships among classes leads to a critical challenge we term *intra-super-class imbalance*. This problem arises from the combination of two factors, best illustrated by *truck* and *automobile* on CIFAR10-LT. First, for classes with **high semantic similarity**, LA-based adjustment offers no remedy because its offset is derived solely from class frequencies and is agnostic to inter-class similarities that drive confusion. Second, the issue is amplified by **extreme local imbalance** within the shared *vehicle* super-class, where *automobile*

---

[*]Equal contribution.

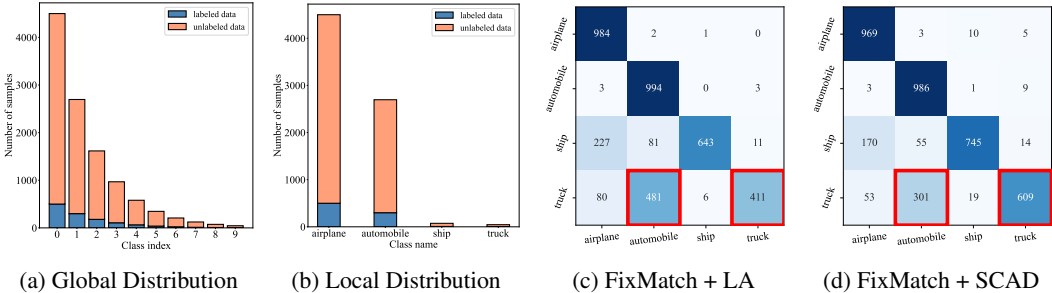

(a) Global Distribution    (b) Local Distribution    (c) FixMatch + LA    (d) FixMatch + SCAD

Figure 1: **Illustration of the *intra-super-class imbalance* problem on CIFAR10-LT.** (a) The overall dataset exhibits a long-tailed distribution. (b) The problem arises from a combination of high semantic similarity and extreme local imbalance, as seen within the *vehicle* super-class. (c) This highlights a limitation of LA (Menon et al., 2021), which struggles to distinguish minority classes from their majority neighbors. (d) In contrast, SCAD mitigates these critical misclassifications.

vastly outnumbers *truck* (Figure 1b). As a result, LA's globally derived correction is fundamentally mismatched to this local conflict. Its boost for *truck* is set by global rarity rather than the intense competition it faces from *automobile*, and thus fails to counteract the dominant class. In pseudo-labeling, this dual failure triggers a self-reinforcing loop of bias amplification, leading to representation overshadowing and the systematic misclassification shown in Figure 1c.

To tackle this problem, we propose **S**uper-**C**lass-**A**ware **D**ebiasing **(SCAD)**, a framework that performs dynamic, super-class-aware logit adjustment. Our key insight is that debiasing should adapt to each sample's predicted super-class, tailoring the correction to the local set of semantically confusable classes. SCAD operationalizes this idea by (i) discovering a latent super-class structure from class names using pre-trained text encoders (e.g., CLIP (Radford et al., 2021)), (ii) training a super-class classifier to infer the semantic context for each unlabeled sample, and (iii) refining standard LA with a super-class-aware correction that focuses the adjustment on classes that compete within the predicted super-class, weighted by the super-class posterior. As a result, SCAD mitigates systematic misclassifications on CIFAR10-LT (Figure 1d) and alleviates analogous *intra-super-class* confusions on more complex long-tailed benchmarks (Figure 4, Appendix A.3).

To validate our approach, we conduct extensive experiments across LTSSL benchmarks, ranging from CIFAR-10/100-LT to large-scale and fine-grained datasets such as ImageNet-127 and Food-101-LT. We show that SCAD is a plug-and-play framework that consistently improves several state-of-the-art methods under distribution-mismatch settings (e.g., prior shift between labeled and unlabeled data). For example, when combined with ACR on CIFAR-100-LT under severe distribution mismatch, SCAD improves accuracy by up to 3.4%. On ImageNet-127, SCAD surpasses prior methods by up to 5%, showing that SCAD mitigates label-scarcity-induced representation challenges, including *intra-super-class* confusions.

## 2  RELATED WORK

**Long-tailed Semi-supervised Learning**  LTSSL addresses the realistic scenario where labeled data is both scarce and imbalanced. Early methods focused on directly manipulating pseudo-labels, for instance, by iterative re-balancing (Wei et al., 2021) or refining their distributions (Kim et al., 2020; Wang et al., 2022). Subsequently, logit adjustment (LA) (Menon et al., 2021) emerged as the *de facto standard* for debiasing, offering a simple correction by directly adjusting the model's output logits. The effectiveness and simplicity of LA have made it the foundational block for recent methods that address distribution mismatch between labeled and unlabeled data. In this setting, the pseudo-label prior can deviate substantially from the labeled prior. These methods, such as ACR (Wei & Gan, 2023) and CDMAD (Lee & Kim, 2024), dynamically modulate the strength of the LA correction. ACR achieves this by estimating the mismatch between labeled and unlabeled distributions, while CDMAD does so by measuring the classifier's intrinsic bias. However, their adjustment schemes remain fundamentally that of LA. Consequently, they are inherently hierarchy-agnostic and sub-optimal for alleviating the localized confusion between semantic neighbors.

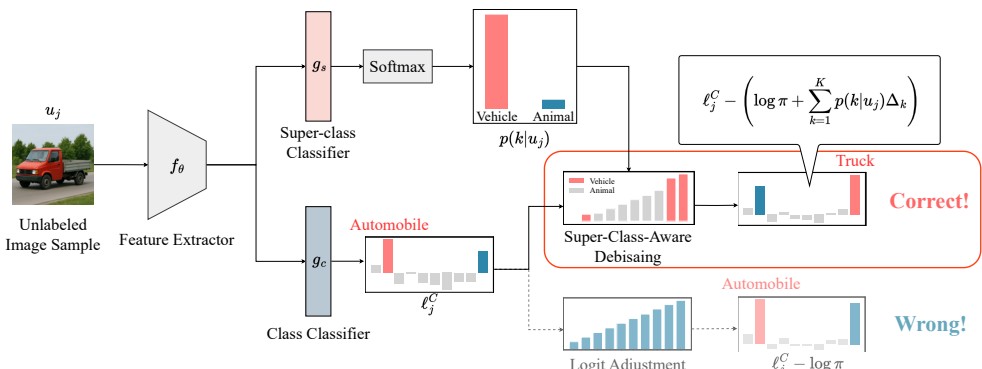

Figure 2: **Illustration of Super-Class-Aware Debiasing (SCAD) mechanism.** For an unlabeled *truck* image, the class classifier initially produces biased logits, incorrectly favoring *automobile*. Standard LA applies a uniform, global correction, which is insufficient to change the outcome. In contrast, our mechanism uses a super-class classifier to infer the sample's context (*vehicle*). In this context, $p(k|u_j)$ is used to compute a targeted local adjustment. When this local adjustment is added to the standard LA, it helps overcome the initial bias and boosts the logit for the correct class.

**Hierarchical and Super-class learning**    Leveraging class hierarchies to improve recognition is well established. In supervised learning, methods like HD-CNN (Yan et al., 2015) and Deep-RTC (Wu et al., 2020) use manually defined hierarchies, while SuperDisco (Du et al., 2023) discovers super-classes from image features via a graph neural network. However, these approaches are less practical for LTSSL, as they often require predefined taxonomies or computationally intensive training. Additionally, the resulting hierarchies can be sensitive to the representation used to construct them, which may be unreliable in the low-label regime. In contrast, SCAD is designed to be lightweight and practical: it constructs super-classes by applying a pre-trained text encoder to class names, without manual hierarchies or additional vision pretraining. This makes SCAD easy to integrate into existing LTSSL pipelines with minimal engineering effort. It then serves as a simple, plug-and-play debiasing module for existing LTSSL methods.

## 3    METHOD

We propose Super-Class-Aware Debiasing (SCAD), a multi-stage framework to address *intra–super-class imbalance* in LTSSL. SCAD consists of three components: (i) super-class discovery from semantic information, (ii) a dedicated training procedure for the feature extractor and two classifiers, and (iii) a dynamic logit adjustment mechanism that corrects biased predictions at inference time. We detail each component below, after introducing notation and problem setup.

### 3.1    PRELIMINARIES

#### 3.1.1    PROBLEM SETUP

Let $\mathcal{D}^l = \{(x_i, y_i)\}_{i=1}^N$ be a labeled dataset and $\mathcal{D}^u = \{u_j\}_{j=1}^M$ an unlabeled dataset, where $x_i, u_j \in \mathbb{R}^d$ denote the $d$-dimensional inputs. We consider $C$ classes, indexed by $c \in [C] \triangleq \{1, \dots, C\}$. Let $N_c$ be the number of labeled samples in class $c$ and $M_c$ the (generally unknown) number of unlabeled samples in class $c$. The labeled set is long-tailed with $N_1 \geq N_2 \geq \cdots \geq N_C$ and imbalance ratio $\gamma_l = N_1/N_C$. We similarly define the unlabeled imbalance ratio $\gamma_u = \max_c M_c / \min_c M_c$.

Our model consists of three main components: a feature extractor $f_\theta : \mathbb{R}^d \to \mathbb{R}^p$, a primary class classifier $g_c : \mathbb{R}^p \to \mathbb{R}^C$, and an auxiliary super-class classifier $g_s : \mathbb{R}^p \to \mathbb{R}^K$, where $p$ is the dimensionality of the feature representation and $K$ denotes the number of discovered super-classes. For a given input $x$, we denote the extracted features as $z = f_\theta(x)$, the class logits as $\ell^c = g_c(z)$, and the super-class logits as $\ell^s = g_s(z)$.

### 3.1.2 SEMI-SUPERVISED LEARNING.

Our method builds upon the consistency regularization paradigm, as popularized by FixMatch (Sohn et al., 2020). The total loss for the primary SSL is

$$L \;=\; L_s \;+\; L_u, \tag{1}$$

where $L_s$ is the supervised loss on labeled data and $L_u$ is the unsupervised consistency loss on unlabeled data.

**Supervised loss.**   For a mini-batch $\mathcal{B}_l \subset \mathcal{D}^l$, we apply a weak augmentation $\mathcal{A}_\mathrm{w}$ to $x_i$ and compute

$$L_s \;=\; \frac{1}{|\mathcal{B}_l|} \sum_{(x_i, y_i) \in \mathcal{B}_l} \mathrm{CE}\big(\mathrm{softmax}\big(g_c(f_\theta(\mathcal{A}_\mathrm{w}(x_i)))\big), \, y_i\big), \tag{2}$$

where $\mathrm{CE}(\cdot, \cdot)$ denotes the standard cross-entropy with the label $y_i$ treated as a class index.

**Unsupervised consistency loss.**   For a mini-batch $\mathcal{B}_u \subset \mathcal{D}^u$, we obtain a pseudo-label from a weak view and enforce consistency on a strong view. Let $q_j = \mathrm{softmax}\big(g_c(f_\theta(\mathcal{A}_\mathrm{w}(u_j)))\big)$ and $\hat{y}_j = \arg\max_{c \in [C]} (q_j)_c$. We apply a strong augmentation $\mathcal{A}_\mathrm{s}$ to the same sample and compute

$$L_u \;=\; \frac{1}{|\mathcal{B}_u|} \sum_{u_j \in \mathcal{B}_u} \mathbb{I}(\max_c (q_j)_c \geq \tau) \; \mathrm{CE}\big(\mathrm{softmax}\big(g_c(f_\theta(\mathcal{A}_\mathrm{s}(u_j)))\big), \, \hat{y}_j\big), \tag{3}$$

where $\tau \in (0, 1)$ is a confidence threshold. The mask $\mathbb{I}(\max_c (q_j)_c \geq \tau)$ prevents low-confidence pseudo-labels from contributing to the loss. In practice, $\hat{y}_j$ is treated with stop-gradient.

## 3.2 SUPER-CLASS GENERATION

The foundation of our method is a structural prior that groups fine-grained classes into meaningful super-classes. We automatically generate this prior from class-name semantics, requiring no manual annotation or pre-trained vision models. More details are explained in A.8.

**Generation Process.**   The generation process involves two steps: (1) We convert the names of all $C$ classes into semantic vectors using a pre-trained text encoder (e.g., SBERT (Reimers & Gurevych, 2019) or CLIP's text encoder (Radford et al., 2021)). (2) We then apply agglomerative clustering on these vectors. This bottom-up technique builds a dendrogram representing the class hierarchy. By cutting this dendrogram at a level that yields a pre-specified number of clusters ($K$), we partition the $C$ classes into $K$ super-classes. This process yields a deterministic mapping $\mathcal{M} : \{1, \dots, C\} \to \{1, \dots, K\}$, where $\mathcal{M}(c)$ is the super-class index for class $c$.

**Justification.**   Our approach of generating super-classes is grounded in the observation that semantic relationships captured by pre-trained text embeddings often correlate with visual taxonomies (Radford et al., 2021). This strategy is particularly advantageous as it sidesteps the need for manual hierarchy creation, which is often infeasible for large-scale or specialized datasets. It is important to clarify that the novelty of our work does not lie in this generation process itself, which is a straightforward application of existing tools. Rather, we employ this process as a simple and effective means to obtain the structural prior required for our main contribution: a new, dynamic debiasing mechanism. Our goal is to show that even an approximate hierarchy can alleviate critical issues when leveraged by a powerful debiasing algorithm.

Crucially, the effectiveness of our framework does not hinge on any single text encoder. This robustness is a key finding, as it shows that our method's principle is general and not reliant on a specific, potentially expensive text model. SCAD consistently improves performance regardless of the text encoder used, from lightweight GloVe (Pennington et al., 2014) to large-scale CLIP. This indicates that SCAD's benefit stems from the debiasing mechanism rather than encoder capacity, a conclusion further supported by our ablations in Section 4.5.

In this work, we obtain super-classes by applying text encoders to class names, but SCAD itself is agnostic to the hierarchy source: it only requires a mapping from classes to super-classes. The same interface can be instantiated using existing taxonomies or manually defined hierarchies. In extremely rare or highly domain-specific settings where class names provide a limited semantic signal, ontology- or expert-defined super-classes offer a natural alternative to text-based grouping.

### 3.3 TRAINING PROCEDURE

The feature extractor $f_\theta$, class classifier $g_c$, and super-class classifier $g_s$ are trained jointly. The goal of this stage is not only to train the main classifier, but also to train a reliable super-class classifier $g_s$ that can provide a trustworthy signal for our debiasing mechanism.

The rationale for this joint training hinges on the observation that the auxiliary task of super-class classification is inherently simpler and more robust to data imbalance. This robustness stems from two factors: (1) it is a coarse-grained problem with significantly fewer classes ($K \ll C$), and (2) grouping classes often mitigates the label distribution imbalance, as noted in prior work (Du et al., 2023). Consequently, $g_s$ can learn to produce reliable predictions even for minority class samples, providing a stable signal that is crucial for our debiasing mechanism. As we show in Appendix A.4, $g_s$ indeed yields a more balanced and reliable prediction distribution on real-world unlabeled data.

To achieve this, we introduce an auxiliary SSL loss for the super-class task, $L_{\text{super}}$, which mirrors the structure of the primary loss:

$$L_{\text{super}} = L_s^{\text{super}} + L_u^{\text{super}}. \tag{4}$$

Here, the supervised component, $L_s^{\text{super}}$, is the standard cross-entropy loss computed on the labeled batch $\mathcal{B}_l$ using super-class targets $y_i^{\text{super}} = \mathcal{M}(y_i)$, where $\mathcal{M}$ maps each fine-grained class to a super-class (either from a dataset-provided hierarchy or from our automatically discovered grouping). The unsupervised component, $L_u^{\text{super}}$, is a consistency loss computed on the unlabeled batch $\mathcal{B}_u$, enforcing that the model's super-class predictions are consistent across strong and weak augmentations.

The final training objective is a weighted sum of the primary and auxiliary losses:

$$L_{\text{total}} = L + \lambda L_{\text{super}}, \tag{5}$$

where $\lambda$ is a hyperparameter balancing the two tasks. This objective encourages $f_\theta$ to learn representations that are discriminative at both fine-grained and coarse-grained levels.

### 3.4 SUPER-CLASS–AWARE LOGIT ADJUSTMENT

This section introduces our framework's core contribution: a novel logit adjustment method, Super-Class–Aware logit Adjustment (SCAD), that addresses a critical limitation of the standard LA. Standard LA applies a static, global correction to the logits based on the class prior distribution $\pi$:

$$\ell_j^{\text{LA}} = \ell_j^c - \log \pi, \tag{6}$$

While effective against global class imbalance, this super-class-aware correction is ill-equipped to mitigate local ambiguities, particularly conflicts arising between classes within the same super-class.

SCAD overcomes this limitation with a dynamic, sample-aware adjustment mechanism, illustrated in Figure 2. The process begins by inferring the coarse-grained context for each unlabeled sample $u_j$. We utilize the super-class classifier, $g_s$, to obtain a posterior probability distribution over super-classes, $p(k|u_j) = \text{softmax}(\ell_j^s)_k$. This distribution, $p(k|u_j)$, quantifies the estimated likelihood of the sample belonging to each super-class $k$. For instance, an image of a *truck* would yield a high probability for the *vehicle* super-class, i.e., $p(\text{vehicle}|u_j) \approx 1$.

This inferred context is then used to apply a tailored, super-class-aware adjustment. We pre-calculate an adjustment vector $\Delta_k \in \mathbb{R}^C$ for each super-class $k$ to counteract *intra-super-class imbalance* within that group. Let $\mathcal{C}_k = \{c \mid \mathcal{M}(c) = k\}$ be the set of all classes belonging to super-class $k$. The $c$-th component of $\Delta_k$ is defined as:

$$(\Delta_k)_c = \begin{cases} \beta_{k,c}, & \text{if } c \in \mathcal{C}_k, \\ \max_{c' \in \mathcal{C}_k} \beta_{k,c'}, & \text{if } c \notin \mathcal{C}_k, \end{cases} \tag{7}$$

where $\beta_{k,c} = n_{k,c}/(\max_{c' \in \mathcal{C}_k} n_{k,c'})$ is the relative dominance score of class $c$ within its super-class $k$, based on the sample count or estimated frequency $n_{k,c}$. We estimate $n_{k,c}$ from a mixture of labeled and high-confidence pseudo-labeled samples: whenever a labeled example with ground-truth label $c$, or an unlabeled example whose pseudo-label is $c$ with confidence above the FixMatch threshold $\tau$, is assigned to super-class $k$, we increment $n_{k,c}$. These counts are recomputed periodically (once per epoch in our implementation) using the current classifier. The normalization in Eq. 7

Table 1: Results on CIFAR10-LT and CIFAR100-LT datasets for various algorithms and settings.

| | CIFAR10-LT | | | | CIFAR100-LT | | | |
| | $\gamma = \gamma_l = \gamma_u = 100$ | | $\gamma = \gamma_l = \gamma_u = 150$ | | $\gamma = \gamma_l = \gamma_u = 10$ | | $\gamma = \gamma_l = \gamma_u = 20$ | |
| Algorithm | $N_1 = 500$ $M_1 = 4000$ | $N_1 = 1500$ $M_1 = 3000$ | $N_1 = 500$ $M_1 = 4000$ | $N_1 = 1500$ $M_1 = 3000$ | $N_1 = 50$ $M_1 = 400$ | $N_1 = 150$ $M_1 = 300$ | $N_1 = 50$ $M_1 = 400$ | $N_1 = 150$ $M_1 = 300$ |
|---|---|---|---|---|---|---|---|---|
| Supervised | $47.3 \pm 0.95$ | $61.9 \pm 0.41$ | $44.2 \pm 0.33$ | $58.2 \pm 0.29$ | $29.6 \pm 0.57$ | $46.9 \pm 0.22$ | $25.1 \pm 1.14$ | $41.2 \pm 0.15$ |
| w/ LA | $53.3 \pm 0.44$ | $70.6 \pm 0.21$ | $49.5 \pm 0.40$ | $67.1 \pm 0.78$ | $30.2 \pm 0.44$ | $48.7 \pm 0.89$ | $26.5 \pm 1.31$ | $44.1 \pm 0.42$ |
| FixMatch | $67.8 \pm 1.13$ | $77.5 \pm 1.32$ | $62.9 \pm 0.36$ | $72.4 \pm 1.03$ | $45.2 \pm 0.55$ | $56.5 \pm 0.06$ | $40.0 \pm 0.96$ | $50.7 \pm 0.25$ |
| w/ DARP | $74.5 \pm 0.78$ | $77.8 \pm 0.63$ | $67.2 \pm 0.32$ | $73.6 \pm 0.73$ | $49.4 \pm 0.20$ | $58.1 \pm 0.44$ | $43.4 \pm 0.87$ | $52.2 \pm 0.66$ |
| w/ CReST+ | $76.3 \pm 0.86$ | $78.1 \pm 0.42$ | $67.5 \pm 0.45$ | $73.7 \pm 0.34$ | $44.5 \pm 0.94$ | $57.4 \pm 0.18$ | $40.1 \pm 1.28$ | $52.1 \pm 0.21$ |
| w/ DASO | $76.0 \pm 0.37$ | $79.1 \pm 0.75$ | $70.1 \pm 1.81$ | $75.1 \pm 0.77$ | $49.8 \pm 0.24$ | $59.2 \pm 0.35$ | $43.6 \pm 0.09$ | $52.9 \pm 0.42$ |
| w/ DASO + Ours | $75.1 \pm 0.10$ | $79.2 \pm 1.60$ | $69.0 \pm 0.61$ | $75.8 \pm 0.33$ | $50.0 \pm 0.22$ | $58.7 \pm 0.28$ | $44.5 \pm 0.35$ | $53.2 \pm 0.50$ |
| FixMatch + LA | $75.3 \pm 2.45$ | $82.0 \pm 0.36$ | $67.0 \pm 2.49$ | $78.0 \pm 0.91$ | $47.3 \pm 0.42$ | $58.6 \pm 0.36$ | $41.4 \pm 0.93$ | $53.4 \pm 0.32$ |
| w/ DARP | $76.6 \pm 0.92$ | $80.8 \pm 0.62$ | $68.2 \pm 0.94$ | $76.7 \pm 1.13$ | $50.5 \pm 0.78$ | $59.9 \pm 0.32$ | $44.4 \pm 0.65$ | $53.8 \pm 0.43$ |
| w/ CReST+ | $76.7 \pm 1.13$ | $81.1 \pm 0.57$ | $70.9 \pm 1.18$ | $77.9 \pm 0.71$ | $44.0 \pm 0.21$ | $57.1 \pm 0.55$ | $40.6 \pm 0.55$ | $52.3 \pm 0.20$ |
| w/ DASO | $77.9 \pm 0.88$ | $82.5 \pm 0.08$ | $70.1 \pm 1.68$ | $79.0 \pm 2.23$ | $50.7 \pm 0.51$ | $60.6 \pm 0.71$ | $44.1 \pm 0.61$ | $55.1 \pm 0.72$ |
| w/ DASO + Ours | $81.6 \pm 0.22$ | $84.0 \pm 0.99$ | $74.5 \pm 1.21$ | $82.2 \pm 0.14$ | $51.8 \pm 0.28$ | $60.5 \pm 0.21$ | $45.7 \pm 0.56$ | $55.7 \pm 0.57$ |
| FixMatch + ACR | $81.6 \pm 0.19$ | $84.1 \pm 0.39$ | $77.0 \pm 1.19$ | $80.9 \pm 0.22$ | $51.3 \pm 0.48$ | $61.1 \pm 0.11$ | $44.8 \pm 0.21$ | $55.9 \pm 0.31$ |
| w/ Ours | $\mathbf{83.5} \pm 0.16$ | $\mathbf{85.5} \pm 0.03$ | $\mathbf{78.6} \pm 0.56$ | $\mathbf{83.3} \pm 0.20$ | $\mathbf{52.7} \pm 0.11$ | $\mathbf{61.8} \pm 0.21$ | $\mathbf{45.8} \pm 0.20$ | $\mathbf{56.4} \pm 0.10$ |

Table 2: Test accuracy of previous LTSSL algorithms and ours under inconsistent class distributions, i.e., $\gamma_l \neq \gamma_u$, on CIFAR-10-LT and STL10-LT datasets. The $\gamma_l$ is fixed to 100 for CIFAR-10-LT, while it is set to 10 and 20 for STL10-LT dataset. The best results are in **bold**.

| | CIFAR10-LT ($\gamma_l \neq \gamma_u$) | | | | STL10-LT ($\gamma_u = N/A$) | | | |
| | $\gamma_u = 1$ (uniform) | | $\gamma_u = 1/100$ (reversed) | | $\gamma_l = 10$ | | $\gamma_l = 20$ | |
| Algorithm | $N_1 = 500$ $M_1 = 4000$ | $N_1 = 1500$ $M_1 = 3000$ | $N_1 = 500$ $M_1 = 4000$ | $N_1 = 1500$ $M_1 = 3000$ | $N_1 = 150$ $M_1 = 100k$ | $N_1 = 450$ $M_1 = 100k$ | $N_1 = 150$ $M_1 = 100k$ | $N_1 = 450$ $M_1 = 100k$ |
|---|---|---|---|---|---|---|---|---|
| FixMatch | $73.0 \pm 3.81$ | $81.5 \pm 1.15$ | $62.5 \pm 0.94$ | $71.8 \pm 1.70$ | $56.1 \pm 2.32$ | $72.4 \pm 0.71$ | $47.6 \pm 4.87$ | $64.0 \pm 2.27$ |
| w/ DARP | $82.5 \pm 0.75$ | $84.6 \pm 0.34$ | $70.1 \pm 0.22$ | $80.0 \pm 0.93$ | $66.9 \pm 1.66$ | $75.6 \pm 0.45$ | $59.9 \pm 2.17$ | $72.3 \pm 0.60$ |
| w/ CReST | $83.2 \pm 1.67$ | $87.1 \pm 0.28$ | $70.7 \pm 2.02$ | $80.8 \pm 0.39$ | $61.7 \pm 2.51$ | $71.6 \pm 1.17$ | $57.1 \pm 3.67$ | $68.6 \pm 0.88$ |
| w/ CReST+ | $82.2 \pm 1.53$ | $86.4 \pm 0.42$ | $62.9 \pm 1.39$ | $72.9 \pm 2.00$ | $61.2 \pm 1.27$ | $71.5 \pm 0.96$ | $56.0 \pm 3.19$ | $68.5 \pm 1.88$ |
| w/ DASO | $86.6 \pm 0.84$ | $88.8 \pm 0.59$ | $71.0 \pm 0.95$ | $80.3 \pm 0.65$ | $70.0 \pm 1.19$ | $78.4 \pm 0.80$ | $65.7 \pm 1.78$ | $75.3 \pm 0.44$ |
| FixMatch + ACR | $92.1 \pm 0.18$ | $\mathbf{93.5} \pm 0.11$ | $85.0 \pm 0.09$ | $89.5 \pm 0.17$ | $77.1 \pm 1.24$ | $83.0 \pm 0.32$ | $75.1 \pm 0.70$ | $81.5 \pm 0.25$ |
| w/ Ours | $\mathbf{93.0} \pm 0.13$ | $93.4 \pm 0.56$ | $\mathbf{86.1} \pm 0.10$ | $\mathbf{89.8} \pm 0.18$ | $\mathbf{77.8} \pm 0.45$ | $\mathbf{83.6} \pm 0.45$ | $\mathbf{75.8} \pm 0.10$ | $\mathbf{82.0} \pm 0.35$ |

yields $\beta_{k,c} \in [0, 1]$, making $(\Delta_k)_c$ a *relative* penalty and rendering the dynamic term insensitive to the overall dataset size. Since the SCAD correction term $\sum_k p(k \mid u_j) \Delta_k$ is a convex combination of bounded adjustments, its components are also bounded in $[0, 1]$ and thus act as a moderate, local correction on top of the global LA term $-\log \pi$, whose magnitude can be much larger on highly imbalanced datasets. We empirically found this construction to be numerically stable across all benchmarks. For the *vehicle* super-class, this formulation assigns a high penalty (up to 1) to the dominant *automobile* class and a smaller penalty to the minority *truck* class. For all classes outside this super-class, a fixed maximum penalty (i.e., the strongest suppression level) is applied to prevent potential confusion. This max-penalty design is motivated by prior work (Tao et al., 2023).

The final SCAD adjustment is formulated by augmenting the standard LA correction with our sample-aware term. Specifically, we use the inferred super-class posterior $p(k|u_j)$ as a dynamic weight to create a convex combination of the pre-computed adjustment vectors, $\Delta_k$. This yields a single, sample-specific adjustment vector. The complete SCAD formulation is:

$$\ell_j^{\text{SCAD}} = \ell_j^c - \left( \log \pi + \sum_{k=1}^{K} p(k|u_j)\Delta_k \right), \tag{8}$$

This dynamic formulation is the key to SCAD's effectiveness. As illustrated in Figure 2, for an image of a truck, the model's high confidence in the *vehicle* super-class ($p(\text{vehicle}|u_j) \approx 1$) ensures that final adjustment is dominated by $\Delta_{\text{vehicle}}$. This vector applies a large penalty to the competing majority class (*automobile*) and a smaller penalty to the true class (*truck*), resolving the local ambiguity. In stark contrast, standard LA applies a fixed, global correction that is blind to this local context, making it ill-equipped to handle such *intra-super-class* conflicts.

Table 3: Test accuracy on CIFAR100-LT under *uniform* and *reversed* distributions.

| Algorithm | $\gamma_u = 1$ (uniform) | | $\gamma_u = 1/10$ (reversed) | |
|---|---|---|---|---|
| | $N_1 = 50$ $M_1 = 400$ | $N_1 = 150$ $M_1 = 300$ | $N_1 = 50$ $M_1 = 400$ | $N_1 = 150$ $M_1 = 300$ |
| FixMatch | $45.5 \pm 0.71$ | $58.1 \pm 0.72$ | $44.2 \pm 0.43$ | $57.3 \pm 0.19$ |
| w/ DARP | $43.5 \pm 0.95$ | $55.9 \pm 0.32$ | $36.9 \pm 0.48$ | $51.8 \pm 0.92$ |
| w/ CReST | $43.5 \pm 0.30$ | $59.2 \pm 0.25$ | $39.0 \pm 1.11$ | $56.4 \pm 0.62$ |
| w/ CReST+ | $43.6 \pm 1.60$ | $58.7 \pm 0.16$ | $39.1 \pm 0.77$ | $56.4 \pm 0.78$ |
| w/ DASO | $53.9 \pm 0.66$ | $61.8 \pm 0.98$ | $\mathbf{51.0} \pm 0.19$ | $60.0 \pm 0.31$ |
| w/ Ours | $\mathbf{54.0} \pm 0.77$ | $\mathbf{62.4} \pm 0.98$ | $48.6 \pm 0.19$ | $\mathbf{60.4} \pm 0.10$ |
| FixMatch + ACR | $57.2 \pm 0.19$ | $66.7 \pm 0.30$ | $51.6 \pm 0.12$ | $62.9 \pm 0.25$ |
| w/ Ours | $\mathbf{59.1} \pm 0.25$ | $\mathbf{66.8} \pm 0.22$ | $\mathbf{53.4} \pm 0.11$ | $\mathbf{63.3} \pm 0.12$ |

Table 4: Comparison with other LTSSL baselines on CIFAR10/100-LT.

| Algorithm | CIFAR-10-LT | | | CIFAR-100-LT |
|---|---|---|---|---|
| | $\gamma_l = \gamma_u = 100$ | $\gamma_l = 100, \gamma_u = 1$ | $\gamma_l = 100, \gamma_u = 1/100$ | $\gamma_l = \gamma_u = 20$ |
| FixMatch + SAW | $77.5 \pm 0.65$ | $\mathbf{81.2 \pm 0.68}$ | $72.3 \pm 0.65$ | $50.1 \pm 0.10$ |
| w/ Ours | $\mathbf{79.4 \pm 0.70}$ | $81.1 \pm 0.30$ | $\mathbf{75.7 \pm 0.57}$ | $\mathbf{53.1 \pm 0.16}$ |
| FixMatch + ABC | $81.1 \pm 1.14$ | $82.7 \pm 0.40$ | $68.9 \pm 0.61$ | $53.3 \pm 0.79$ |
| w/ Ours | $\mathbf{82.0 \pm 0.30}$ | $\mathbf{83.0 \pm 0.42}$ | $\mathbf{73.2 \pm 1.50}$ | $\mathbf{55.1 \pm 0.19}$ |
| FixMatch + CoSSL | $83.1 \pm 0.45$ | $88.8 \pm 0.42$ | $\mathbf{85.1 \pm 0.58}$ | $53.9 \pm 0.78$ |
| w/ Ours | $\mathbf{83.3 \pm 0.32}$ | $\mathbf{89.0 \pm 0.27}$ | $84.8 \pm 0.47$ | $\mathbf{55.0 \pm 0.21}$ |
| FixMatch + CDMAD | $83.6 \pm 0.46$ | $87.5 \pm 0.46$ | $77.6 \pm 0.70$ | $54.3 \pm 0.44$ |
| w/ Ours | $\mathbf{85.1 \pm 0.11}$ | $\mathbf{87.9 \pm 0.10}$ | $\mathbf{79.4 \pm 1.15}$ | $\mathbf{54.9 \pm 0.76}$ |

## 4 EXPERIMENT

We conducted experiments used datasets in LTSSL, including CIFAR10-LT (Krizhevsky et al., 2009), CIFAR100-LT (Krizhevsky et al., 2009), and STL10-LT (Coates et al., 2011). Additionally, we performed experiments on the ImageNet-127, Food101-LT (Fan et al., 2022) dataset.

Followed by Wei & Gan (2023), we used three different settings: *consistent*, *uniform*, and *reverse*. The *consistent* setting refers to when $\gamma_l = \gamma_u$. The *uniform* setting indicates that $\gamma_u = 1$. Finally, the *reverse* setting describes the case where $\gamma_l$ and $\gamma_u$ have reciprocals of the consistent setting.

### 4.1 IMPLEMENTATION DETAILS

Following DASO (Oh et al., 2022) and ACR (Wei & Gan, 2023) settings, we apply our framework to various existing methods, including FixMatch (Sohn et al., 2020), FixMatch+DASO, and FixMatch+ACR. If the base method does not adopt the LA-based pseudo-label debiasing, we apply only super-class learning.

We conducted the experiment for the LTSSL method in different experiment settings including SAW (Lai et al., 2022), ABC (Lee et al., 2021), CoSSL (Fan et al., 2022), and CDMAD (Lee & Kim, 2024) to show that our framework can enhance the other method in a different setting. For detailed experimental settings, please refer to Appendix A.2. We set the top-1 accuracy as our evaluation metric. All experiments were conducted three times with different random seeds. Please note that the results for ACR differ from those reported in the original paper[*]. this discrepancy arises because the authors mistakenly used a different backbone on CIFAR100-LT.

---

[*]https://github.com/Gank0078/ACR/issues/5

Table 5: Test accuracy on ImageNet-127. The best results are in **bold**.

| Algorithm / ImageNet-127 | $32 \times 32$ | $64 \times 64$ |
|---|---|---|
| FixMatch | 29.7 | 42.3 |
| w/ DARP | 30.5 | 42.5 |
| w/ DARP+cRT | 39.7 | 51.0 |
| w/ CReST+ | 32.5 | 44.7 |
| w/ CReST++LA | 40.9 | 55.9 |
| w/ CoSSL | 43.7 | 53.9 |
| w/ ACR | 57.2 | 63.6 |
| w/ Ours | 60.1 | 66.7 |
| w/ ACR + Ours | **60.5** | **67.0** |

## 4.2 RESULTS ON CIFAR10/100-LT AND STL10-LT

**In case of** $\gamma_\ell = \gamma_u$ : To demonstrate the effectiveness of our proposed framework, we conducted experiments by additionally incorporating FixMatch + DASO and FixMatch + ACR for comparison. As shown in Table 1, SCAD enhances performance across almost all settings. It shows that SCAD integrates effectively with various existing LTSSL algorithms, yielding a complementary effect. In particular, the CIFAR100-LT performance was impressive with the ACR, where the settings $N_1 = 50$ and $M_1 = 400$ achieved state-of-the-art performance improvements of 2.7% and 2.2% when $\gamma_\ell = \gamma_u = 10$ and $\gamma_\ell = \gamma_u = 20$.

**In case of** $\gamma_\ell \neq \gamma_u$ : In the real world, there can be distributional differences between unlabeled and labeled datasets. Therefore, we conducted experiments with both uniform and reverse settings to explore these scenarios.

We observed overall performance improvements when SCAD was applied to ACR in the uniform and reverse settings. Specifically, as shown in Table 3, SCAD marginally outperformed DASO on CIFAR100-LT in both the uniform and reverse settings. Furthermore, combining ACR with SCAD led to performance gains of up to 2% and 3.4% in the uniform and reverse settings, respectively.

## 4.3 RESULTS ON IMAGENET-127 AND FOOD101-LT

On ImageNet-127 (Fan et al., 2022), a challenging 127-class subset of ImageNet with a severe long-tail distribution ($\gamma_u \approx 286$), our method shows a distinct advantage. Following the standard setup with 10% labeled data, our approach outperforms existing baselines, as detailed in Table 5. Notably, it surpasses the strong ACR baseline by a substantial margin of 5.0% at 32×32 resolution and 4.0% at 64×64 resolution. This improvement is particularly noteworthy as ACR represents a highly competitive recent method, highlighting the efficacy of our proposed hierarchical debiasing. Furthermore, when our method is combined with ACR, it achieves new state-of-the-art results, underscoring its high compatibility as a plug-and-play module. These significant gains, consistent with results on Food101-LT (see Appendix A.6), validate that our approach effectively alleviates the critical problem of *intra-super-class imbalance*—a fine-grained challenge that existing methods have struggled to address.

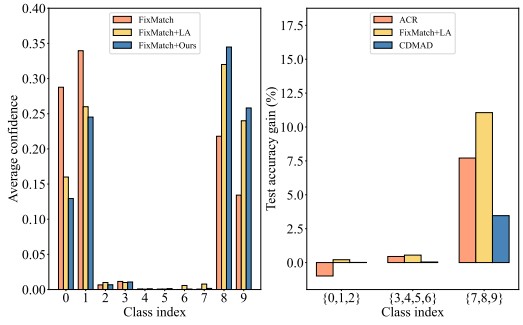

Figure 3: **Left:** figure displays the average confidence scores for minority-class samples (8, 9) of the super-class *vehicle*. Our method effectively reduces confidence in confusing majority classes (0, 1) within the same super-class while increasing confidence in the correct minority classes (8, 9). **Right:** illustration demonstrates that our approach consistently improves accuracy across various LTSSL frameworks, especially benefiting minority classes.

Table 6: Ablation studies of SCAD on CIFAR10-LT and STL10-LT datasets.

| Ablations | CIFAR10-LT | STL10-LT |
|---|---|---|
| FixMatch | 67.8 | 56.1 |
| + Super-class learning | 69.2 | 69.0 |
| + Logit-Adjustment (LA) | 76.9 | 70.4 |
| + Super-class-Aware Debiasing (SCAD) | **78.7** | **71.3** |

Table 7: Comparison of SCAD with ground truth and various text encoders on CIFAR100-LT.

| Algorithms | CIFAR100-LT |
|---|---|
| FixMatch + LA | 47.3 |
| w/ Ours with ground truth | 50.3 |
| w/ Ours with GloVe | 49.7 |
| w/ Ours with SBERT | 50.1 |
| w/ Ours with CLIP text encoder | 50.4 |
| w/ Ours with text-embedding-ada-002 | **50.5** |

## 4.4 RESULTS ON OTHER LTSSL BASELINES

Table 4 shows that our framework can also be applied effectively to other experiment settings with state-of-the-art methods, we conducted experiments on the three settings of CIFAR10-LT and consistent setting of CIFAR100-LT within the CDMAD (Lee & Kim, 2024), CoSSL (Fan et al., 2022), ABC (Lee et al., 2021) and SAW (Lai et al., 2022) framework. According to CDMAD, when we ran experiments with ACR under CDMAD settings, we observed a performance drop from 84.1 to 81.8 at $N_1 = 1500$ and $M_1 = 3000$. Our framework remains robust when combined with existing LTSSL in different settings. According to Table 4, when combining CDMAD with our framework, which is another state-of-the-art method, there was a 1.6% improvement in the consistent setting, a 0.3% increase in the uniform setting, and a 2.3% improvement in the reverse setting. These results demonstrate that our proposed SCAD framework effectively integrates with diverse LTSSL baselines, further validating its utility and applicability.

## 4.5 COMPREHENSIVE ANALYSIS OF OUR METHOD

**Ablation study**  According to Table 6, we evaluated the performance improvements achieved by incrementally adding each component to the algorithms on CIFAR10-LT and STL10-LT under a consistent setting. First, adding the super-class learning component resulted in significant performance gains, with an increase of 2.0% on CIFAR10-LT and 23% on STL10-LT. Finally, adding SCAD yielded additional gains of 9.2% on CIFAR10-LT and 1.1% on STL10-LT. The limited improvement observed from adding SCAD on STL10-LT is likely due to the significant difference in the number of labeled and unlabeled datasets.

**Comparison with ground truth super-class labels and various text embedding models**  As shown in Table 7, there is no performance difference between using our super-class labels generated from text-embeddings-ada-002 (Ryan et al., 2022) or the CLIP text encoder (Radford et al., 2021), and using the ground-truth super-class labels provided in CIFAR100. We further evaluate SCAD on CIFAR100-LT using weaker embeddings such as GloVe (Pennington et al., 2014), SBERT (Reimers & Gurevych, 2019), and still outperform the baseline. This indicates that our approach remains effective even without large-scale models.

**Minority class accuracy comparison**  As shown in Figure 3, Our method improves the average confidence of minority classes (8,9) over the LA baseline, while correspondingly reducing the confidence of majority classes (0,1) within the same super-class. This result yields consistent gains across diverse LTSSL frameworks, highlighting its general applicability. Moreover, since our approach merely appends a super-class classifier trained in a joint manner, it entails negligible computational overhead (more detailed in Appendix A.7).

**Hyperparameter settings** we define the confidence threshold $\tau$ for class-level predictions as $0.95$, following FixMatch. Since super-class learning is expected to converge faster than class-level learning, we set the super-class threshold $\tau_s$ to the same value, i.e., $\tau_s = 0.95$, for simplicity and convenience. We determined the number of super-classes $K$ by setting $K = \lceil \frac{C}{4} \rceil$. For a detailed analysis, choosing $K$ and ensuring super-class balance, please refer to Appendix A.4.

## 5 CONCLUSION

In this paper, we introduce Super-Class-Aware Debiasing (SCAD), a novel framework designed to tackle the fundamental challenge of *intra-super-class imbalance* in long-tailed semi-supervised learning (LTSSL). Unlike previous approaches that rely solely on global class priors for logit adjustment, SCAD dynamically adapts the corrective force to the semantic context of each sample. To validate the effectiveness of our approach, we conduct extensive experiments on a wide range of LTSSL benchmarks. The results demonstrate that SCAD substantially improves the reliability of pseudo-labels, enhances the learning dynamics of minority classes, and consistently narrows the performance gap caused by long-tailed distributions. Moreover, SCAD achieves state-of-the-art results across diverse datasets, outperforming strong baselines and existing debiasing methods. These findings highlight that incorporating semantic structure and local class relationships into the debiasing process is a powerful strategy, and establish SCAD as a new standard for robust and generalizable learning in real-world imbalanced settings.

## 6 LIMITATIONS AND FUTURE WORKS

Our study mainly evaluates SCAD within LTSSL. Its effectiveness for other tasks such as detection or segmentation remains an open question and would require further validation. A promising direction for future work is to extend SCAD beyond classification. In particular, applying our idea to more complex tasks such as object detection or semantic segmentation could further demonstrate its utility, as these tasks often suffer from both long-tailed distributions and semantic overlap among categories.

## 7 ACKNOWLEDGEMENTS

We thank Changhun Kim and Juho Lee for insightful advice and feedback that greatly improved this work. This work was partly supported by the Institute of Information & Communications Technology Planning & Evaluation (IITP) grant funded by the Korea government (MSIT) (No. RS-2022-II220311, Development of Goal-Oriented Reinforcement Learning Techniques for Contact-Rich Robotic Manipulation of Everyday Objects (31%); No. RS-2024-00457882, AI Research Hub Project; No. RS-2019-II190079, Artificial Intelligence Graduate School Program (Korea University)); by the IITP-ITRC (Information Technology Research Center) grant funded by the Korea government (Ministry of Science and ICT) (No. IITP-2025-RS-2024-00436857) (32%); by the BK21 Four project of the National Research Foundation of Korea; by the National Research Foundation of Korea (NRF) grant funded by the Korea government (MSIT) (No. RS-2025-00560367); by the IITP under the Artificial Intelligence Star Fellowship support program (No. IITP-2025-RS-2025-02304828) (32%) funded by the Korea government (MSIT); and by KOREA HYDRO & NUCLEAR POWER CO., LTD (No. 2024-Tech-09).

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

# A APPENDIX

## A.1 THE ASSISTANCE OF A LARGE LANGUAGE MODEL (LLM)

This work used a large language model (LLM) solely for language editing (grammar, wording, and minor typographical corrections) during manuscript preparation.

## A.2 EXPERIMENT SETTING

For a fair comparison, we integrate our framework with the DASO (Oh et al., 2022), ACR (Wei & Gan, 2023), SAW (Lai et al., 2022), ABC (Lai et al., 2022), CoSSL (Fan et al., 2022), and CDMAD (Lee & Kim, 2024) algorithms using their official code repositories. For example, we use the code for DASO from `https://github.com/ytaek-oh/daso.git`, for ACR from `https://github.com/Gank0078/ACR.git`, and for CDMAD from `https://github.com/LeeHyuck/CDMAD.git`.

When combining FixMatch (Sohn et al., 2020) with our framework, we adopt the ACR settings. It is worth noting that the settings for each algorithm exhibit slight differences. For instance, ACR and CDMAD utilize distinct configurations. The ACR setting is specifically tailored to generate weakly and strongly augmented versions of input images for use within the FixMatch framework.

- **Weak augmentation:** This pipeline applies random horizontal flipping and random cropping with reflective padding to preserve the image structure. These transformations are designed to introduce minimal changes to the input image while maintaining its semantic content.

- **Strong augmentation:** In addition to the weak augmentations, this pipeline uses the RandAugment (Cubuk et al., 2020) method, which applies 2 transformations with 10 magnitude with Cutout (DeVries & Taylor., 2017) operation.

In CDMAD setting, Weak augmentation is same however, there are slight difference in Strong augmentation. CDMAD uses RandAugment with 3 transformations of magnitude 4, followed by the Cutout operation.

The hyperparameter settings differ significantly between ACR and CDMAD. In ACR, the labeled dataset batch size is set to 64, the ratio of the unlabeled batch size to the labeled batch size is set to 1, and the optimizer's learning rate is adjusted using a cosine decay schedule. The learning rate at each step is calculated as $\alpha \cos\left(\frac{7\pi t}{16T}\right)$, where $\alpha$ is the initial learning rate set to $3 \times 10^{-3}$, $t$ denotes the current iteration and $T$ is the total number of iterations.

In contrast, CDMAD uses a labeled dataset batch size of 32, the ratio of the unlabeled batch size to the labeled batch size is set to 2, and a learning rate of $2 \times 10^{-3}$. All experiments are conducted using one NVIDIA RTX 3090 GPU. The software environment includes PyTorch 2.1.2, TorchVision 0.16.2, and CUDA 12.1.

All settings, We performed the experiments on CIFAR10-LT (Krizhevsky et al., 2009), CIFAR100-LT (Krizhevsky et al., 2009), and STL-10 (Coates et al., 2011) using the WideResNet-28-2 (Zagoruyko & Komodakis, 2016) architecture, and on ImageNet-127 (Fan et al., 2022), Food101-LT(Fan et al., 2022) using the ResNet-50 (He et al., 2016) architecture.

## A.3 INTRA-SUPERCLASS-IMBALANCE PROBLEM ON CIFAR100-LT

We extended our investigation to CIFAR100-LT to prove the prevalence of intra-super-class imbalance. Our results (see Figure 4) show that even on CIFAR-100-LT, substantial imbalance persists within super-classes when using standard methods such as LA: minority classes inside each semantic group remain underserved despite global prior correction. In contrast, SCAD consistently reduces these intra–super-class disparities by explicitly targeting the local bias. These additional findings suggest that intra–super-class imbalance is not a CIFAR10-specific artifact and that our solution is robust across different long-tailed datasets.

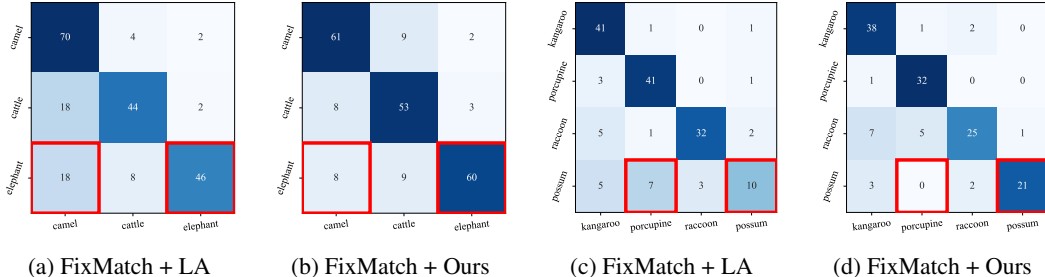

| (a) FixMatch + LA | (b) FixMatch + Ours | (c) FixMatch + LA | (d) FixMatch + Ours |

Figure 4: Illustration of the *intra-super-class imbalance* problem on CIFAR100-LT. Within the Figure 4a, 4b super-class (e.g., camel and elephant), the model frequently confuses semantically related classes, and SCAD effectively alleviates these errors. A similar pattern appears in the Figure 4c, 4d super-class (e.g., porcupine and possum), where SCAD reduces misclassifications by addressing the local dominance within the group.

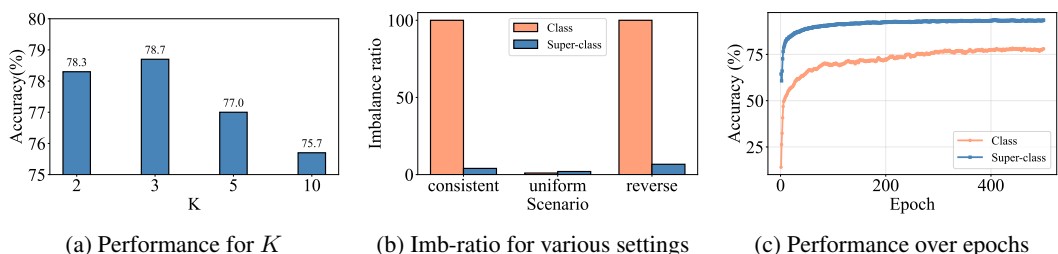

| (a) Performance for $K$ | (b) Imb-ratio for various settings | (c) Performance over epochs |

Figure 5: Illustration for balancing super-classes and determining the optimal value of $K$ on CIFAR10-LT

### A.4 ANALYSIS OF SUPER-CLASS LEVEL DISTRIBUTION

According to Figure 5a, we set $K = \lceil \frac{C}{4} \rceil$. In addition, in Figure 5b we compare the class-level and superclass-level imbalance ratios of unlabeled data under the consistent, uniform, and reverse settings. Note that in the reverse setting, the reciprocal values are not employed. We observe that the imbalance ratio drops sharply in the consistent and reverse settings, while it slightly increases in the uniform setting but still remains nearly identical. Furthermore, owing to this more balanced distribution structure, Figure 5c reports class and super-class level accuracies across training epochs with our method, consistently achieves higher performance at all epochs. These results indicate that the super-class classifier is more balanced and reliable than the class-level counterpart. Experiments on CIFAR100-LT show a similar trend: According to Figure 6a, the performance reaches its maximum around $K = \lceil \frac{C}{4} \rceil$, and when the original unlabeled imbalance ratio $\gamma_u$ is 10 or lower, the performance difference becomes negligible. (See Figure 6b)

### A.5 VISUALIZATION ON SCAD

To visually demonstrate that our proposed method is particularly beneficial for minority classes, we further compare the confusion matrices of FixMatch, FixMatch + Ours, ACR and ACR+Ours. Specifically, as shown in Figure 7, our method significantly enhances classification performance for minority classes under consistent settings, highlighting the effectiveness of SCAD in alleviating class imbalance.

### A.6 FOOD101-LT RESULTS

Food101-LT, we sample 250 labeled and 500 unlabeled images per class, corresponding to imbalance ratios of $\gamma_l = \gamma_u = 50, 100$, followed by CoSSL. According to Table 8, Our method also improves the accuracy on Food101-LT about 3% and 7%, respectively.

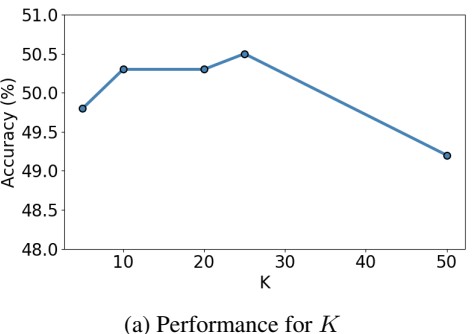 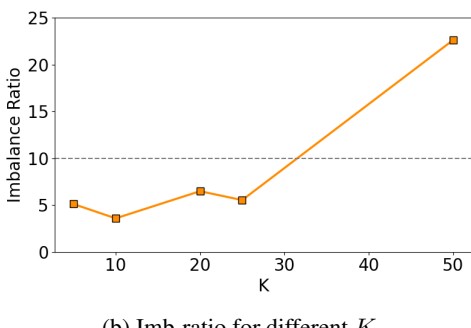

(a) Performance for $K$          (b) Imb-ratio for different $K$

Figure 6: Illustration for balancing super-classes and determining the optimal value of $K$ on CIFAR100-LT

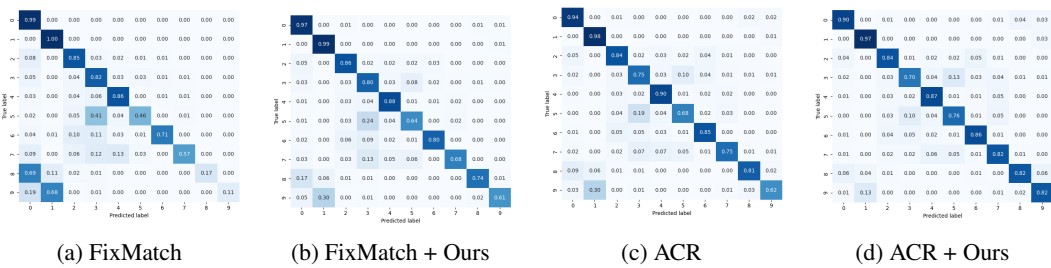

(a) FixMatch      (b) FixMatch + Ours      (c) ACR      (d) ACR + Ours

Figure 7: Confusion matrices of the test set for FixMatch, FixMatch+Ours, ACR, and ACR+Ours under consistent settings.

## A.7 CALIBRATION ANALYSIS ON SCAD

We further evaluate the calibration properties of SCAD compared to FixMatch and LA baselines. We measure calibration using Expected Calibration Error (ECE), which quantifies the discrepancy between predicted confidence and empirical accuracy.FixMatch exhibits severe miscalibration in tail classes, with ECE values reaching up to 0.59, while SCAD reduces these errors substantially. This indicates that SCAD alleviates underconfidence in tail classes and achieves a more balanced calibration across head and tail. We futher evaluated calibration by integrating SCAD into ACR and observed consistent improvements across all frequency groups. The ECE drops from 0.1018 to 0.0517 (Head), 0.1339 to 0.1228 (Medium), and from 0.1654 to 0.0926 (Tail). These results show that SCAD not only preserves performance but also substantially enhances prediction reliability, especially for tail classes.

Training time analysis As presented in Table 10, the reported time represents the training time per epoch. The difference between our method and the baseline is approximately 1 second, which is negligible in practice.

## A.8 DETAILED INFORMATION ABOUT THE SUPER-CLASS

We retrieved descriptions for each class name using the Wikipedia API, and for convenience, we only used the first sentence of the returned summary. If no corresponding Wikipedia entry exists, we simply use the class name itself. Agglomerative clustering is performed using the clustering module from scikit-learn. As previously mentioned, we set $K = \left\lceil \frac{C}{4} \right\rceil$. Specifically, for CIFAR10-LT (Krizhevsky et al., 2009) and STL10-LT (Coates et al., 2011), $K = 3$, for CIFAR100-LT (Krizhevsky et al., 2009), $K = 25$ and for ImageNet-127 (Fan et al., 2022), $K = 32$. The groupings for CIFAR10-LT, STL10-LT, CIFAR100-LT and ImageNet-127 are detailed in Table 11 and Table 13. These groupings are determined using the text-embedding-ada-002 model (Ryan et al., 2022).

Table 8: Test accuracy on Food101-LT with $\gamma_l = \gamma_u = 50, 100$. The best results are in **bold**.

| Algorithm | $\gamma = 50$ | $\gamma = 100$ |
|---|---|---|
| FixMatch | 42.6 | 35.3 |
| w/ DARP | 42.0 | 34.2 |
| w/ DARP + cRT | 41.5 | 34.4 |
| w/ CReST+ | 43.8 | 31.2 |
| w/ CReST+ + LA | 47.7 | 36.1 |
| w/ CoSSL | 49.0 | 40.4 |
| w/ Ours | **51.0** | **43.5** |

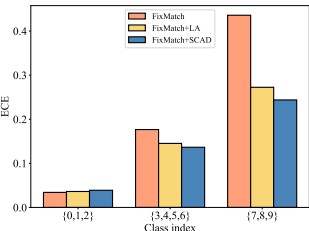

Figure 8: ECE analysis on CIFAR10-LT

Table 9: ECE comparison between ACR and ACR+Ours

| Method | Head | Medium | Tail |
|---|---|---|---|
| ACR | 0.1018 | 0.1339 | 0.1654 |
| ACR+Ours | **0.0517** | **0.1228** | **0.0926** |

Table 10: Training time comparison (seconds) per epoch

| Method | FixMatch | FixMatch+Ours | ACR | ACR+Ours |
|---|---|---|---|---|
| Time (s) | 28 | 29 | 35 | 36 |

Table 11: Super-class table on CIFAR10-LT. We use the Wikipedia API to retrieve a summary for a given query and perform agglomerative clustering using the text-embedding-ada-002 model, setting the number of clusters to 3.

| Cluster | Classes |
|---|---|
| Cluster 0 | 0: airplane, 1: automobile, 8: ship, 9: truck |
| Cluster 1 | 2: bird, 6: frog |
| Cluster 2 | 3: cat, 4: deer, 5: dog, 7: horse |

Table 12: Super-class table on CIFAR100-LT. We use the Wikipedia API to retrieve a summary for a given query and perform agglomerative clustering using the text-embedding-ada-002 model, setting the number of clusters to 25.

| Cluster | Classes |
|---|---|
| Cluster 0 | 1: aquarium_fish, 26: crab, 45: lobster, 77: snail, 99: worm |
| Cluster 1 | 27: crocodile, 29: dinosaur, 44: lizard, 73: shark, 78: snake, 93: turtle |
| Cluster 2 | 21: chimpanzee, 42: leopard, 43: lion, 88: tiger |
| Cluster 3 | 54: orchid, 62: poppy, 70: rose, 82: sunflower, 92: tulip |
| Cluster 4 | 58: pickup_truck, 69: rocket, 85: tank, 89: tractor |
| Cluster 5 | 2: baby, 11: boy, 35: girl, 46: man, 98: woman |
| Cluster 6 | 22: clock, 86: telephone, 87: television |
| Cluster 7 | 6: bee, 7: beetle, 14: butterfly, 18: caterpillar, 24: cockroach, 79: spider |
| Cluster 8 | 36: hamster, 50: mouse, 65: rabbit, 74: shrew, 75: skunk, 80: squirrel |
| Cluster 9 | 0: apple, 51: mushroom, 53: orange, 57: pear, 83: sweet_pepper |
| Cluster 10 | 5: bed, 20: chair, 25: couch, 94: wardrobe |
| Cluster 11 | 23: cloud, 71: sea |
| Cluster 12 | 33: forest, 49: mountain, 60: plain |
| Cluster 13 | 9: bottle, 10: bowl, 28: cup |
| Cluster 14 | 13: bus, 81: streetcar, 90: train |
| Cluster 15 | 15: camel, 19: cattle, 31: elephant |
| Cluster 16 | 17: castle, 37: house, 76: skyscraper |
| Cluster 17 | 16: can, 39: keyboard, 40: lamp, 61: plate, 84: table |
| Cluster 18 | 32: flatfish, 67: ray, 91: trout |
| Cluster 19 | 30: dolphin, 95: whale, 72: seal |
| Cluster 20 | 8: bicycle, 41: lawn_mower, 48: motorcycle |
| Cluster 21 | 3: bear, 4: beaver, 34: fox, 55: otter, 97: wolf |
| Cluster 22 | 12: bridge, 68: road |
| Cluster 23 | 38: kangaroo, 63: porcupine, 66: raccoon, 64: possum |
| Cluster 24 | 47: maple_tree, 52: oak_tree, 56: palm_tree, 59: pine_tree, 96: willow_tree |

Table 13: Super-class table on ImageNet-127. We use the Wikipedia API to retrieve a summary for a given query and perform agglomerative clustering using the text-embedding-ada-002 model, setting the number of clusters to 32.

| Cluster | Classes |
|---|---|
| Cluster 0 | 17: electronic_equipment, 43: light, 60: camera, 99: magazine, 125: monitor |
| Cluster 1 | 16: furniture, 58: litter, 118: cushion |
| Cluster 2 | 2: jewelry, 22: hairpiece, 36: protective_garment, 44: garment, 45: nightwear, 50: uniform, 55: glove, 84: hosiery, 90: gown, 113: dress |
| Cluster 3 | 27: pot, 28: dish, 30: rod, 42: home_appliance, 71: toiletry, 80: pan, 94: cleaning_implement, 106: cooker, 109: turner |
| Cluster 4 | 29: athlete, 34: game_equipment, 41: golf_equipment, 47: gymnastic_apparatus, 53: sports_implement, 110: participant, 115: diver |
| Cluster 5 | 31: frozen_dessert, 37: sauce, 39: dip, 63: pudding, 74: entree, 77: cracker, 111: concoction |
| Cluster 6 | 9: platform, 32: weight, 59: system, 76: structure, 105: sign, 121: drill_rig |
| Cluster 7 | 19: plaything |
| Cluster 8 | 75: ceratopsian, 86: mammal, 104: bird |
| Cluster 9 | 25: bus, 46: fare, 56: sled, 98: passenger_train |
| Cluster 10 | 38: ray, 79: plectognath, 107: shark, 108: ganoid, 124: food_fish, 126: soft_finned_fish |
| Cluster 11 | 72: wine, 81: punch, 114: coffee |
| Cluster 12 | 8: globule, 21: stick, 66: tissue, 83: covering |
| Cluster 13 | 11: fungus, 14: coelenterate, 54: echinoderm, 65: arthropod, 68: mollusk |
| Cluster 14 | 88: rescue_equipment, 91: safety_belt |
| Cluster 15 | 52: spacecraft, 64: rocket, 67: heavier_than_air_craft, 112: lighter_than_air_craft |
| Cluster 16 | 18: sailing_vessel, 35: ship, 93: oar, 119: boat |
| Cluster 17 | 6: anguid_lizard, 24: teiid_lizard, 49: lacertid_lizard, 78: gecko, 87: agamid, 102: venomous_lizard, 116: iguanid, 122: chameleon |
| Cluster 18 | 20: board, 120: bar |
| Cluster 19 | 95: bath_linen, 103: fabric |
| Cluster 20 | 1: vegetable, 4: fruit, 40: fodder, 51: flower |
| Cluster 21 | 70: pen, 101: eraser |
| Cluster 22 | 3: ligament |
| Cluster 23 | 13: frog, 89: snake, 96: turtle, 97: salamander, 100: worm |
| Cluster 24 | 69: cistern, 117: container |
| Cluster 25 | 0: loaf_of_bread, 73: bun |
| Cluster 26 | 5: cap, 26: helmet, 123: hat |
| Cluster 27 | 15: alligator, 62: crocodile |
| Cluster 28 | 12: device, 23: tool, 48: sharpener, 57: kit, 85: key, 92: duplicator |
| Cluster 29 | 10: military_vehicle |
| Cluster 30 | 33: geological_formation |
| Cluster 31 | 7: damselfish, 61: butterfly_fish, 82: scorpaenid |

