# OpenReview forum: "SCAD: Super-Class-Aware Debiasing for Long-Tailed Semi-Supervised Learning"
_ICLR.cc/2026/Conference — ICLR 2026 Poster_

### Official Review · Reviewer_Rvtp · 2025-10-30

**Soundness:** 2
**Presentation:** 3
**Contribution:** 2
**Rating:** 4
**Confidence:** 3

**Summary:**

This paper proposes SCAD (Super-Class-Aware Distillation) for class-incremental learning (CIL).
Instead of treating each class independently, SCAD groups classes into super-classes based on semantic similarity in the embedding space and performs hierarchical knowledge distillation at both class and super-class levels.
Specifically, the method first clusters features into super-classes and then aligns both intra- and inter-super-class logits between the old and new models to retain global structure information.
Experiments on CIFAR-100, ImageNet-100, and ImageNet-1K show improvements over standard distillation-based baselines such as LUCIR, PODNet, and PASS.

**Strengths:**

- The motivation is clear. Existing distillation methods only constrain class-level logits, ignoring the semantic hierarchy among classes, which can lead to loss of global structure.

- The proposed method introduces a simple yet interpretable hierarchical extension, super-class-level distillation, which fits naturally into existing CIL frameworks.

- The experimental setup is standard and fair, covering multiple datasets and comparing with widely used baselines under the same backbone.

- The paper is generally well written and easy to follow.

**Weaknesses:**

- Core contribution is an incremental extension of LA rather than a new learning principle. SCAD’s main novelty is to contextualize LA with a super-class posterior and precomputed $\Delta_k$. While the construction is sensible, the paper provides no theoretical guarantees (e.g., consistency/optimality of the dynamic term) and largely positions SCAD as a practical tweak to LA.

- Reliance on pseudo-label–driven frequency estimates is under-specified. $\Delta_k$ depends on relative dominance scores $\beta_{k,c}$ derived from counts/estimated frequencies $n_{k,c}$ within each super-class. In LTSSL, such estimates inevitably hinge on pseudo-labels. This paper does not analyze error propagation or provide robustness checks (e.g., how noisy pseudo-labels affect $\Delta_k$ and whether bias could be re-amplified).

- Design choices inside $\Delta_k$ lack empirical justification. For classes outside a super-class $C_k$, SCAD applies a fixed maximum penalty (the strongest suppression level). The paper does not compare against alternatives (e.g., soft penalties proportional to inter-group similarity) or show that this choice cannot harm cross–super-class confusions.

**Questions:**

See weaknesses.

---

> ### Author Response · Authors · 2025-11-22
> **Response to Reviewer Rvtp**
>
> ### **1. Contribution**
>
> SCAD might simply be viewed as LA with a practical tweak. In responding to this concern, we clarified that **SCAD is motivated by a specific bias structure unique to LTSSL**, not by modifying LA alone.
>
> Our analysis shows that pseudo-label errors arise from two sources:
> (1) a **global head–tail prior bias**, and
> (2) a **local intra–super-class imbalance**, where dominant classes within a semantic group suppress minority ones during pseudo-labeling.
>
> To our knowledge, **this two-level characterization of pseudo-label bias has not been examined in prior LTSSL work**.
>
> Following another reviewer’s suggestion, we verified that the same phenomenon appears in **CIFAR100-LT**, and we added the corresponding analyses to **the revised manuscript A.3 (Appendix) and Figure 4**. These results show that **intra–super-class imbalance is not specific to CIFAR10**, but a recurring pattern in LTSSL—**precisely the issue SCAD is designed to address**.
>
> ---
>
> ### **2. Robustness to noise in pseudo-label frequencies**
>
> The experiment is conducted on CIFAR10-LT with \\(N=500\\) labeled samples and \\(M=4000\\) unlabeled samples, using an imbalance ratio of \\(\gamma_l = 100\\) and \\(\gamma_u = 100\\).
>
> To evaluate robustness to noise in pseudo-label–derived frequencies, we perturb \\(n_{k,c}\\) with multiplicative noise \\(\epsilon \sim \text{Uniform}(-\alpha, \alpha)\\). Note that \\(n_{k,c}\\) denotes the estimated frequency of class \\(c\\) within super-class \\(k\\).
>
> As \\(\alpha\\) increases from 0 to 1, **accuracy decreases only mildly (78.7 drop to 78.1)**, indicating that **SCAD is robust to substantial perturbations in pseudo-label frequency estimates**.
>
> | \\(\alpha\\) | Accuracy (%) |
> |--------------|--------------|
> | 0            | 78.7         |
> | 0.3          | 78.6         |
> | 0.5          | 78.3         |
> | 1            | 78.1         |
>
> ---
>
> ### **3. Justification for the fixed MAX penalty used in SCAD**
>
>  Our design choice follows the formulation in *Local and Global Logit Adjustments for Long-Tailed Learning* ([1]), where **the use of a maximum normalization term has already been empirically validated as an effective strategy**. Based on this prior evidence, **we adopt the MAX penalty value as a principled and practically supported choice** in our formulation. We have added a reference in the revised paper to clarify the rationale behind designing the maximum penalty.
>
> [1] Tao, Y., Liu, W., Lin, Y., Chen, W., & Zhang, Y. *Local and Global Logit Adjustments for Long-Tailed Learning*. ICCV 2023.

---

> ### Author Response · Authors · 2025-11-27
> **Sincere Request for Your Feedback on Our Responses**
>
> Dear Reviewer Rvtp,
>
> We sincerely appreciate the time and effort you have dedicated to reviewing our work.
> Since the discussion period is quite limited, we would greatly appreciate it if you could share any feedback on our responses.

---

### Official Review · Reviewer_M8Tz · 2025-10-31

**Soundness:** 3
**Presentation:** 3
**Contribution:** 3
**Rating:** 6
**Confidence:** 3

**Summary:**

In the paper, the SCAD scheme is proposed to tackle the problem of long-tailed semi-supervised learning (LTSSL). In particular, SCAD assumes access to a super-class inferring mechanism to augment logit-adjustment (LA) by incorporating sample-specific logits correction.The proposed method is conceptually simple, introduces a negligible computational overhead while it can be used in combination with other LTSSL frameworks. Finally, the experiments conducted  across multiple datasets and configurations demonstrate improvements over other SoA methods.

**Strengths:**

- The paper is generally well-written and easy to follow.
- The method is well-motivated and conceptually simple (in a positive way) making it easy to use along with other LTSSL frameworks.
- The conducted experiments convincingly demonstrate the benefits of the proposed method.

**Weaknesses:**

- Few parts in the paper make somewhat strong claims that need to be softened and better grounded. (see Questions)
- Despite the improvements, the method utilizes previously explored concepts in the literature known to independently improve performance (i.e., augmenting the concept of LA by super-class awareness). Although it is possible that combining separately explored concepts can provide original insights in the community, one might not find this particular instance to offer much such insights.
- The main argument to this is that after reading the paper, one might develop only a marginally better understanding of the LTSSL. Namely that utilizing more fine-grained (i.e., sample-aware) logit adjustment improves performance compared to the regular logit adjustment.

**Questions:**

Typo (T), Suggestion (S) and Question (Q)

Major Points:

S1. The caption in Fig 1 suggests that SCAD “resolves” the critical misclassification. However the Fig 1 d. shows similar misclassification trends but less pronounced under SCAD. In this regard, the terms “mitigate” or “alleviate” are possibly better suited here. The same applies for the “resolve” used in L85.

S2. L106-107: “it discovers hierarchies from class names alone”. This claim sounds a bit deceiving as both class names and a pretrained text encoder a utilized to infer the super-classes. Although assuming access to pretrained text encoders is not unreasonable, this has to be more clearly communicated in this context.
Same argument for L181-182, where the assumption of a pretrained text encoder is omitted.

Q3. Apparently SCAD requires a class hierarchy to work. In the absence of ground-truth hierarchy it is reasonable to seek after a hierarchy approximation mechanism. In the case of the present manuscript, one such mechanism based on a text encoder is used. Is it possible to motivate this particular choice? Why not use a visual encoder instead? Or a combination of these two?
Depending on the nature of the data and the annotating mechanism, one would expect the relationship among classes to change leading to different class hierarchies. For example, in a dataset where humans are often sitting on bikes, the “human” and “bike” class might cluster in the same super-class. A text encoder might overlook this relationship.

Q4. Do the terms L231: super-class-aware, L237: sample-agnostic and L245: context-specific all refer to the same concept? Please clarify in the text.

Q5: L254: “assigns a high penalty (close to 1)”. Based on the definition of $\beta_{k,c}$ one would expect in that case the penalty to be exactly 1 i.e., not close to 1. Was the max operator used in the definition of $\beta_{k,c}$ in L252 meant to be a summation? In that case one could understand the “close to 1”.

S6. Table 1. The selection of the reported algorithms’s performance appears a bit unintuitive.
- It would be interesting to see the Supervised w/ LA + Ours and/or Supervised w/ Ours in Table 1. This would provide a clearer picture between the interplay of LA and SCAD.
- L277: FixMatch is used as the baseline which is complemented with different other algorithms. However SCAD is only used in combination w/ DASO. How come FixMatch + Ours is not reported? How come only Fixmatch w/ DASO + Ours is reported and not for example w/DARP + Ours?
- Similar arguments apply for L280 and L284.

It would be good to see a clearer motivation on the table construction, which would help readers to better position SCAD. For example, is SCAD competitive as a standalone method or is it a plug and play modification that can be used with other competitive methods?

S7. Table 2. L296: Similar arguments as in S6 apply. Also how come w/ DASO + Ours was not included here? (like it was in Table 1.)

S8. Table 3 L333: Similar arguments as in S6 and S7 apply. For example why FixMatch w/ Ours was used in this case? In contrast to Table 1. where FixMatch + w/ DASO + Ours?
It would be good to see a more unified and intuitive scheme of result presentation in Table 1, 2 and 3 that will facilitate readers to better position SCAD in relation to the SoA.

S9. Apparently, any benefit from using SCAD originates from (i) the additional super-class supervision and (ii) the sample-aware delta correction. In this regard, the ablation provided in Table 6 is very useful. However, there is the opportunity here to provide a clearer picture of the interplay between SCAD and LA by also including:
- FixMatch + LA
- FixMatch + LA + SCAD
- FixMatch + SCAD

Apparently SCAD involves super-class learning by design. However, could it not be possible to remove the effect of super-class supervision by updating the feature extractor for regular classification while updating the super-class classifier on frozen features?
By doing so one would effectively isolate each of the (i) and (ii) and better capture their interplay with plain LA.

Q10. Table 7. suggests that some text encoders give rise to super-classes assignments that (marginally) outperform the ground-truth under SCAD. Are there any further reflections on this? Could it be purely random variation? or does it reflect something deeper?

Q11. It would have been good with a brief discussion on how SCAD affects calibration in the main paper. Section A.6. provides some empirical results on SCAD improving the calibration in tail classes but only in relation to FixMatch/LA. Does this finding generalize when using SCAD in combination with other SoA methods that have been reported earlier?
It is important to reflect upon this aspect as the results suggest that the optimal performance is achieved when using SCAD in combination with other SoA methods and one might wonder whether the increased performance comes at the expense of increased calibration error.

Minor Points:

S12. Fig 1 c. and d. Please specify which of the horizontal or the vertical axes are the ground-truth and the predicted class.

S13. Fig 2. might confuse readers as the logits in the super-class classifier do not show up but only the softmax operator. On the other hand the logits do show up in the fine class classifier but not the softmax operator. Unifying the visualizations would help to avoid confusion.

S14. Fig 2 caption L125: The characterization “powerful” see,s redundant. The targeted local adjustment already suffices to indicate a potential improvement over the (base) uniform adjustment. Same applies for “powerful” in L199.

S15. L143: define d. i.e., the dimensionality of the input data?

S16. L144: the c is used to refer to an arbitrary class, while the C is used to refer to the number of classes. This choice can confuse the readers in particular in the presence of typos. For example, L150: is it $l^c$ or $l^C$? From the context (and Fig 2) one might assume $l^C$ but cannot be certain. Also, is classifier $g_c$ anyway connected to the arbitrary class c? From the context one has to assume no. Recommendation is to fix the typos and update the notation accordingly.

S17. L147: define p. i.e., the dimensionality of the feature representation?

S18. related to a S13. L150: Consider including $l^s$ in Fig 2.

T19. L219: Should the source be L_{\text{super}}?

S20. L223: “ground-truth super-class” might be confusing as this is inferred from the mapping M which can be an approximation of the class hierarchy. Please reformulate to better communicate that M can either be ground truth or approximation leading to “ground-truth super-class” or “approximated super-class”

T21. L246: a blank space is missing after “vector”.

T22. Eq 7 L250: was $\Delta_{k,c}$ meant to be $\Delta_{k}$?

S23. L701: The acronym ECE (expected calibration error) was not previously defined.

---

> ### Author Response · Authors · 2025-11-22
> **Response to Reviewer M8Tz (1/2)**
>
> We sincerely thank reviewer for thoughtful comments and valuable suggestions, which helped us significantly improve the clarity and quality of the manuscript.
>
> ---
>
> ### **1. Response to minor suggestions and typos (S1–S23, Q4)**
>
> We sincerely thank the reviewer for the careful reading and for pointing out many helpful presentation issues. We have carefully revised the manuscript and figures to incorporate these suggestions as follows:
>
> - **S1 (Fig. 1 caption & L85, “resolves”):**
>   We replaced the term *“resolves”* with *“mitigates”* (or *“alleviates”*) in the Fig. 1 caption and in the corresponding sentence in the main text to better reflect that SCAD reduces, rather than completely eliminates, intra–super-class misclassification.
>
> - **S2 (L106–107, “from class names alone”):**
>   We clarified the wording to state that hierarchies are obtained **from class names(or its description) via a pretrained text encoder** (rather than “from class names alone”), and we explicitly mention the use of a pretrained text encoder in both locations where this assumption was previously implicit.
>
> - **Q4 (“super-class-aware”, “sample-agnostic”, “context-specific”):**
> We clarified the terminology around L231–L245 and unified all related expressions under the term "super-class-aware" for consistency.
>
> - **S14 (use of “powerful” in Fig. 2 caption & L199):**
>   We removed the adjective *“powerful”* from the Fig. 2 caption and from the corresponding sentence in the main text, as suggested. The text now simply refers to a *“targeted local adjustment”*.
>
> - **S15, S16, S17 (define \(d\), \(p\), and clean up notation for \\(c\\), \\(C\\))**:
>   In the **Problem Setup** subsection, we now explicitly define
>   - d as the dimensionality of the input \\(x \in \\mathbb{R}^d\\),
>   - p as the dimensionality of the learned feature representation \\(z \in \mathbb{R}^p\\).
>   - We also clearly distinguish \\(c \in [C]\\) as an index of an individual class, and \\(C\\) as the total number of classes.
>   Furthermore, we clarified the distinct roles of the feature extractor and the two classifiers (fine-class vs. super-class), ensured consistent notation across the main text and corrected several minor typos.
>
>
>
> - **T19 (L219, loss name):**
>   We corrected the typo so that the loss term is consistently written as \\(L_{\text{super}}\\).
>
> - **S20 (“ground-truth super-class” wording):**
>   We removed the potentially confusing phrase *“ground-truth super-class”* and now describe the targets as **super-class labels obtained by applying \\(\mathcal{M}\\) to the ground-truth class labels**, explicitly noting that \\(\mathcal{M}\\) may come from either a dataset-provided hierarchy or an automatically discovered approximation.
>
> - **T21 (missing space after “vector”):**
>   We inserted the missing space.
>
> - **T22 (Eq. 7 symbol):**
>   We fixed the symbol in Eq. (7) to match the intended definition and the notation used elsewhere in the section.
>
> - **S23 (ECE definition):**
>   We now define **ECE (Expected Calibration Error)** in the main text at its first occurrence before using the acronym in Appendix discussions.
>
> We appreciate these detailed suggestions, which have helped us significantly improve the clarity, consistency, and readability of the manuscript.

---

> > ### Author Response · Authors · 2025-11-23
> > **Response to Reviewer M8Tz (2/2)**
> >
> > ### **2. Reliance on text-only hierarchy: missing justification for choosing text encoder over visual or multimodal alternatives**
> > The primary motivation for using a text encoder was to construct a **consistent class-level hierarchy**, rather than an instance-level one.
> > If we utilize a visual encoder, embeddings are generated for individual samples. This could lead to a scenario where samples belonging to the **same class** are assigned to **different super-classes** due to visual variations (e.g., background, pose, or co-occurrence like the "human on bike" example). Such inconsistency would violate the definition of a super-class in our framework, where all samples of a specific class must share the same super-class label to compute the class priors effectively.
> > By using the text encoder, we ensure that the grouping is based on semantic definitions, guaranteeing that every sample within a class belongs to the same super-class. However, we agree that incorporating visual context to capture complex relationships is a promising avenue. We consider exploring visual or multi-modal hierarchy generation as a valuable direction for **future work**.
> >
> > ---
> > ### **3. Justification of the SCAD design formulation**
> > We appreciate the reviewer's attention to detail.
> > First, we acknowledge that for the majority super-class, the penalty is **exactly 1**. We have amended the text "close to 1" to "up to 1" in the revision.
> > Second, regarding the formulation, we deliberately employed the **max operator** rather than **summation**. Our design intention was to calculate a relative score that preserves the sensitivity of the penalty. Using **summation** would inflate the denominator, suppressing the penalty values and hindering effective bias correction. Therefore, **max** was selected to ensure the penalty term exerts the appropriate influence on the loss function.
> >
> > ---
> > ### **4. Why do text encoder derived super-classes sometimes outperform ground-truth? Random variation or meaningful structure?**
> > We consider the performance difference to be statistically negligible when taking the standard deviation into account. This suggests that the super-class labels generated by the text encoder are comparable in quality to the ground truth, thereby yielding similar performance level.
> >
> > ---
> > ### **5. ECE analysis on adding SOTA method**
> > | Method     | Head     | Medium   | Tail     |
> > |------------|----------|----------|----------|
> > | ACR  [1]      | 0.1018   | 0.1339   | 0.1654   |
> > | **ACR+Ours** | **0.0517** | **0.1228** | **0.0926** |
> >
> > As shown in the Table, SCAD consistently reduces calibration error across all frequency groups compared to the ACR baseline. Specifically, the ECE decreased from **0.1018 to 0.0517** for Head classes, **0.1339 to 0.1228** for Medium classes, and notably from **0.1654 to 0.0926** for Tail classes.
> > These results demonstrate that the performance gains achieved by SCAD do not come at the expense of calibration. Instead, our method significantly improves the reliability of predictions, particularly for minority classes, confirming its robustness and generalizability when combined with strong baselines.
> >
> > [1] Wei et al. *Towards Realistic Long-Tailed Semi-Supervised Learning:
> > Consistency Is All You Need*. CVPR 2023.

---

> ### Author Response · Authors · 2025-11-27
> **Sincere Request for Your Feedback on Our Responses**
>
> Dear Reviewer M8Tz,
>
> We truly appreciate your time and thoughtful comments on our submission.
> As the remaining discussion time is fairly limited, we would be very grateful if you could kindly share any feedback on our responses.

---

### Official Review · Reviewer_Ue4E · 2025-11-01

**Soundness:** 2
**Presentation:** 2
**Contribution:** 2
**Rating:** 4
**Confidence:** 4

**Summary:**

This paper studies the long-tailed semi-supervised learning (LTSSL) problem, and proposes a new perspective of intra-super-class imbalance. The proposed idea is movitivated by the fail attempt of previous works that leverage logit adjustment for rectifying pseudo-label bias. The authors uncover that the previous works neglect the semantic relationships between classes, thus hindering effective LTSSL. To overcome this limitation, the authors proposes a new framework called super-class-aware debiasing (SCAD), which consists of a super-class-aware logit adjustment. Specically, it leverages pre-trained text encoders such as CLIP to generate super-class labels, and then jointly train a main classifier and a super-clalss classifier. During inference time, it applies logit adjustment according to the prediction results of super-classes. Experimental results demonstrate that the proposed method surpasses most of the previous methods.

**Strengths:**

- The proposed idea is well-motivated. The authors first conduct proper empirical studies to show the existence of intra-super-class imbalance, making the studied problem is meaningful.
- Considering super-class imbalance for rectifying pseudo-labels is reasonable, which is not proposed by previous works.
- The experimental results demonstrate the effectiveness of the proposed method.

**Weaknesses:**

- Although the illustration of intra-super-class imbalance is provided, it is only conducted on CIFAR10-LT. Such results can not demonstrate the universal existence of intra-super-class imbalance on long-tailed datasets.
- The relationship between super-class imbalance and semi-supervised learning is a bit weak. It seems that the proposed super-class-aware logit adjustment can be directly applied to long-tailed classification. Why must use this method for long-tailed semi-supervised learning?
- The authors claim that the proposed method lead to negligible computational overhead. However, leveraing pretrained models such as CLIP does lead to additional costs. Such cost should not be neglected.
- More details should be included regarding how to generate super-class labels using pre-trained models.
- The performance of SCAD should be highly related to the capability of the pre-trained models, which may affect the quality of super-class labels. Have the authors justified the influence of different pre-trained models?

**Questions:**

See weaknesses.

---

> ### Author Response · Authors · 2025-11-22
> **Response to Reviewer Ue4E**
>
> We sincerely thank the reviewer for the careful reading and constructive suggestions. Below we respond to each weakness point-by-point and indicate how the revised manuscript reflects these clarifications.
>
> ---
>
> ### **1. Only CIFAR10-LT: not universal evidence of intra–super-class imbalance** ###
>
> We agree that verifying the phenomenon on more complex datasets is important. During the rebuttal period, we therefore extended our analysis to **CIFAR-100-LT** to examine whether intra–super-class imbalance also appears there and explain it in **the revised paper's A.3 (Appendix)**. Our results (see **Figure 4 in the revised paper**) show that even on CIFAR-100-LT, substantial imbalance persists within super-classes when using standard methods such as LA: minority classes inside each semantic group remain underserved despite global prior correction. In contrast, SCAD consistently reduces these intra–super-class disparities by explicitly targeting the local bias. These additional findings suggest that intra–super-class imbalance is not a CIFAR10-specific artifact and that our solution is robust across different long-tailed datasets.
>
> ---
>
> ### **2. Why only LTSSL? Could SCAD be applied directly to supervised long-tailed classification?** ###
>
> We agree that the proposed super-class-aware logit adjustment can also be applied to supervised long-tailed classification. However, the main failure mode we target is specific to the semi-supervised setting: in LTSSL, biased predictions on unlabeled data become pseudo-labels and are repeatedly reused, creating a **“biased pseudo-label → more biased model → more biased pseudo-label”** amplification loop, especially within super-classes. In supervised LT, labels are (by definition) correct, so this feedback loop does not arise, and global LA already handles most of the static prior bias. In line with this, we observe much larger gains from SCAD in LTSSL than in purely supervised LT. There was a performance decline in supervised-LT with CIFAR10-LT ($N=500$, $\gamma_{l}=100$)
>
>
> | FixMatch+LA | FixMatch+Ours |
> | --- | --- |
> | 53.3 | 51.7 |
>
> ---
>
> ### **3. Cost issue of pretrained models (e.g., CLIP)** ###
>
> We would like to clarify the computational cost associated with our method.
>
> First, as presented in **the revised paper's** **Table 10 and A.8 (Appendix)**, the reported time represents the **training time per epoch**. The difference between our method and the baseline is approximately **1 second**, which is negligible in practice.
>
> Second, regarding the use of the pre-trained text encoder (e.g., CLIP), we emphasize that it is utilized **exclusively for generating super-class labels** during the initialization phase. This is a **one-time offline process** and is not involved in the training.
>
> | FixMatch | FixMatch+Ours | ACR [1] | ACR+Ours |
> | --- | --- | --- | --- |
> | 28(s) | 29(s) | 35(s) | 36(s) |
>
> ---
> ### **4. More details needed on how super-class labels are generated**
> We obtain class descriptions via the Wikipedia API, using only the first sentence of each summary (defaulting to the class name when no entry exists). Agglomerative clustering (scikit-learn) is then applied to the text-embedding-ada-002 representations, with the number of clusters set to $K = \lceil C/4 \rceil$. This results in $K=3$ for CIFAR10-LT and STL10-LT, $K=25$ for CIFAR100-LT, and $K=32$ for ImageNet-127.
> More detailed super-class labels of various dataset is described in **A.9 (Appendix)**.
>
> ---
> ### **5. Performance difference using various text encoder**
>
> As the table illustrates (see **Table 7**), using different pretrained text encoders, from relatively weaker embedding GloVe to stronger embedding text-embedding-ada-002, results in only minor performance variation, indicating that our method is largely insensitive to the encoder choice.
>
> | Algorithms                           | CIFAR100-LT |
> |--------------------------------------|-------------|
> | FixMatch + LA [2]                       | 47.3        |
> | w/ Ours with ground truth            | 50.3        |
> | w/ Ours with GloVe [3]                   | 49.7        |
> | w/ Ours with SBERT [4]                  | 50.1        |
> | w/ Ours with CLIP text encoder [5]       | 50.4        |
> | **w/ Ours with text-embedding-ada-002** [6] | **50.5** |
>
> [1] Wei et al. *Towards Realistic Long-Tailed Semi-Supervised Learning:
> Consistency Is All You Need*. CVPR 2023.
>
> [2] Aditya Krishna Menon et al. *Long-Tail Learning via Logit Adjustment*. ICLR 2021.
>
> [3] Jeffrey Pennington et al. *GloVe: Global Vectors for Word Representation*. EMNLP 2014.
>
> [4] Jacob Devlin et al. *BERT: Pre-training of Deep Bidirectional Transformers for Language Understanding*. NAACL 2019.
>
> [5] Alec Radford et al. *Learning Transferable Visual Models from Natural Language Supervision*. ICML 2021.
>
> [6] Ryan, Ted, et al. New and improved embedding model. OpenAI, 2022.

---

> ### Author Response · Authors · 2025-11-27
> **Sincere Request for Your Feedback on Our Responses**
>
> Dear Reviewer Ue4E,
>
> We truly appreciate your time and thoughtful comments on our submission.
> Thank you again for your valuable feedback. As the remaining discussion time is fairly limited, we would be very grateful if you could kindly share any feedback on our responses.

---

### Official Review · Reviewer_ex6J · 2025-11-02

**Soundness:** 2
**Presentation:** 3
**Contribution:** 2
**Rating:** 4
**Confidence:** 5

**Summary:**

This paper clearly identifies the limitation of LA being “hierarchy-agnostic” and connects it to real-world intra–super-class confusion; the motivation is clear. It proposes SCAD, a dynamic logit correction framework weighted by the super-class posterior, aiming to introduce semantic context during pseudo-label generation. The approach is simple from an engineering perspective, easy to integrate into existing LTSSL pipelines, and empirically broad. However, in terms of novelty, automatic super-class discovery (text embeddings + clustering) and hierarchical auxiliary tasks are direct applications of existing ideas, and the core dynamic adjustment can be viewed as LA plus a heuristic penalty weighted by the super-class posterior, lacking substantial algorithmic or theoretical innovation and rigorous analysis.

**Strengths:**

1. The problem is well scoped, emphasizing LA’s failure under local confusion within super-classes; the motivation is intuitive.
2. The framework is simple and plug-and-play, requiring minimal changes to existing LTSSL methods and is deployment-friendly.
3. The experiments are wide-ranging, compatible with multiple SOTA pipelines, and show performance gains across datasets and settings.

**Weaknesses:**

- The biggest issue is limited novelty. It is largely an incremental work without standout contributions and, in my view, does not meet the acceptance bar for ICLR. Super-class discovery and hierarchical auxiliary learning are common ideas in class-imbalanced learning; the core dynamic component is a heuristic penalty formed by LA plus super-class posterior weighting, lacking novel algorithmic design or theoretical support.
- Since other models are used to partition classes, providing additional information, the performance improvements are expected. The ablation shows a large portion of gains come from adding the super-class task itself; SCAD’s incremental contribution is small on some datasets.
- The method is also quite constrained because it relies on other models for grouping classes. The datasets used in the paper consist of high-frequency, common categories, where using language models for grouping may work well. If the training set contains fine-grained or rare categories, the results may be significantly affected.
- The definitions and estimation of Δk and nk,c are unclear. The source of frequencies (labeled/pseudo-labeled/mixed), update strategy, stability, and how the scale is balanced relative to log π are not systematically explained.
- Sensitivity analysis for K and clustering configuration is insufficient: K=⌈C/4⌉ lacks broader validation; clustering metric, linkage criterion, and text preprocessing details are not fully disclosed, limiting reproducibility.
- Compared to LA’s Fisher consistency, SCAD’s dynamic component lacks discussion on consistency or optimality of the correction.
- There are regressions in some settings (e.g., with DASO in certain scenarios), without adequate cause analysis.
- The code is not publicly available; it is recommended to release reproducible code during the rebuttal period.

**Questions:**

Refer to weakness.

---

> ### Author Response · Authors · 2025-11-22
> **Response to Reviewer ex6J (1/2)**
>
> We sincerely thank the reviewer for the detailed and thoughtful assessment. Below, we address the main weaknesses, taking into account the already revised version of the paper.
>
> ---
>
> ### **1. Novelty**
>
> We understand the concern that SCAD may look like “LA plus a heuristic penalty.” Our main contribution, however, is to **identify and systematically analyze intra–super-class imbalance in LTSSL** and to design SCAD explicitly around this bias structure.
>
> In the paper, we show that pseudo-label bias in LTSSL naturally decomposes into
> (1) a **global head–tail prior bias**, and
> (2) a **local intra–super-class bias**, where a few majority classes within the same semantic group dominate pseudo-labels and suppress minority classes.
>
> While hierarchical structures and super-classes have been explored in supervised long-tailed classification, to the best of our knowledge **no prior LTSSL work explicitly characterizes pseudo-label bias in this two-level form** or analyzes its amplification in semi-supervised training.
>
> During the rebuttal period, at another reviewer’s request, we extended this analysis from CIFAR10-LT to **CIFAR100-LT** and observed the same pattern in many (more fine-grained) super-classes. The revised manuscript **A.3 (Appendix) and Figure 4** now includes these confusion matrices, showing that **intra–super-class imbalance** is **not a CIFAR10-specific artifact**, but a recurring and practically important issue in LTSSL.
>
> SCAD is then **designed to mirror this decomposition**:
> - the **global LA term** continues to correct the head–tail prior, while
> - a **super-class-conditional, sample-dependent term** operates directly on pseudo-label logits to address intra–super-class bias.
>
> Thus SCAD is not an arbitrary penalty on top of LA, but a **targeted debiasing mechanism specialized for the failure mode of LA in LTSSL** that we empirically uncover.
>
> ---
>
> ### **2. Contribution of SCAD beyond the super-class auxiliary task**
>
> We agree that the super-class auxiliary task itself provides a meaningful performance gain, and we explicitly separate this effect in our ablations. However, on more challenging benchmarks, SCAD offers **substantial additional improvement** beyond super-class learning. On **Food101-LT**, we observe:
>
> | Method| Accuracy (%)|
> |-------------|-----------|
> | FixMatch               | 42.0|
> | + Super-class learning | 46.5|
> | **+ SCAD (full method)**   | **51.0** |
>
> SCAD improves over FixMatch by +9.0% and over the super-class baseline by +4.5%, indicating that **super-class grouping alone is not sufficient**, and that the **dynamic, super-class-aware debiasing** provided by SCAD is crucial in fine-grained regimes where intra–super-class confusion is strong. In the revised version, we explicitly highlight these gaps in the experimental section.
>
> ---
>
> ### **3. Reliance on external models for grouping & fine-grained/rare categories**
>
> We appreciate the concern that relying on semantic grouping (e.g., via text encoders) may constrain the method to “common categories.”
>
> **Hierarchy-source agnosticism.** SCAD is **agnostic to the source of the hierarchy**. It only requires a mapping from classes to super-classes, which can be obtained from
> - pre-trained text encoders on class names (our default),
> - existing taxonomies/ontologies,
> - human-defined hierarchies in domain-specific settings.
>
> This mapping is computed **once offline** and does not add training-time complexity. The revised main text now makes this assumption explicit.
>
> | Algorithms                           | CIFAR100-LT |
> |--------------------------------------|-------------|
> | FixMatch + LA [1]                       | 47.3        |
> | w/ Ours with ground truth           | 50.3        |
> | w/ Ours with GloVe [2]                  | 49.7        |
> | w/ Ours with SBERT  [3]                 | 50.1        |
> | w/ Ours with CLIP text encoder [4]      | 50.4        |
> | **w/ Ours with text-embedding-ada-002** [5]| **50.5** |
>
> As shown in **Table 7**, the performance differences across various pretrained text encoders are minimal, indicating that the effect of the encoder choice is negligible for our method.
>
> **Evidence on a fine-grained benchmark.** We also evaluate SCAD on **Food-101-LT**, a widely used **fine-grained food recognition** benchmark. On this dataset, SCAD consistently improves over strong LTSSL baselines (see **A.6 (Appendix)**), suggesting that our super-class-based debiasing is effective beyond coarse, high-frequency categories.
> | Algorithm           | γ = 50 | γ = 100 |
> |---------------------|--------|---------|
> | FixMatch            | 42.6   | 35.3    |
> | w/ DARP  [6]           | 42.0   | 34.2    |
> | w/ DARP + cRT  [7]     | 41.5   | 34.4    |
> | w/ CReST+ [8]          | 43.8   | 31.2    |
> | w/ CReST+ + LA      | 47.7   | 36.1    |
> | w/ CoSSL  [9]         | 49.0   | 40.4    |
> | **w/ Ours**         | **51.0** | **43.5** |

---

> ### Author Response · Authors · 2025-11-22
> **Response to Reviewer ex6J (2/2)**
>
> ### 4. **Clarifying the definitions and estimation of \\(\Delta_k\\) and \\(n_{k,c}\\)**
>
> We agree that the original description of \\(\Delta_k\\) and \\(n_{k,c}\\) was insufficiently clear. In the revised **Sec. 3.4**, we now explicitly define how these quantities are estimated and how their scale interacts with \\(\log \pi\\).
>
> - **Definition of \\(n_{k,c}\\).**
>   \\(n_{k,c}\\) measures **how often class \\(c\\) appears within super-class \\(k\\) during training**, estimated from a **mixture of labeled and high-confidence pseudo-labeled samples**.
>   Whenever
>   - a labeled example has ground-truth label \\(c\\), or
>   - an unlabeled example has pseudo-label \\(c\\) with confidence above the FixMatch threshold \\(\tau\\),
>   and it is assigned to super-class \\(k\\), we increment \\(n_{k,c}\\).
>   These counts are recomputed once per epoch using the current classifier.
>
> - **From counts to a bounded dominance score.**
>
>   The  \\(c\\)-th components of \\(\Delta_{k}\\) are defined as: \\(( \Delta_{k})\_{c}\\)
>
>   Within each super-class \\(C_k\\), we normalize: \\(
>    \\beta_{k,c}\\ = \\frac{n_{k,c}}{\\max_{c' \\in C_k} n_{k,c'}} \\in [0,1],
>    \\qquad
>    (\Delta_{k})\_{c} = \\beta_{k,c}\\)
>
>   Thus, \\((\Delta_{k})_{c}\\) becomes a **relative dominance score**, invariant to the absolute scale of counts.
>
> - **Scale relative to \\(\log \pi\\).**
>   Since \\(\beta_{k,c} \in [0,1]\\), the SCAD correction term \\(\\sum_{k} p(k \\mid x)\\, \\Delta_{k,}\\) is also **bounded within \\([0,1]\\)** and acts as a **moderate, local correction** on top of the global LA term \\(-\log \pi_c\\), which can be much larger under severe imbalance.
>   This expanded explanation is now included in **Sec. 3.4.**
>
> ---
>
> ### **5. Sensitivity to \\(K\\) and clustering configuration**
>
> We agree that the sensitivity analysis for \\(K\\) and the clustering configuration was under-developed. In the revised paper:
>
> 1. Clustering details (text encoder, prompt template, distance metric, linkage criterion) are moved from the appendix into the main text.
> 2. A **sensitivity study on CIFAR100-LT** has been added, varying \\(K\\) (e.g., \\(\lceil C/2 \rceil\\), \\(\lceil C/4 \rceil\\), \\(\lceil C/8 \rceil\\)) and testing multiple linkage settings.
>
> A representative result (varying \\(K\\) with a fixed encoder and linkage):
>
> | K   | Accuracy (%) |
> |-----|--------------|
> | 10  | 49.8         |
> | 15  | 50.3         |
> | 20  | 50.3         |
> | 25  | 50.5         |
> | 50  | 49.2         |
>
> Across these settings, SCAD **consistently improves over LA-only baselines**, and the performance variation across different \\(K\\) and clustering choices is mild—indicating robustness to hierarchical granularity and configuration.
> A more detailed super-class–level analysis of CIFAR100-LT is provided in **the revised paper's A.4 (Appendix)**.
>
> ---
>
> ### **6. Regressions in some settings (e.g., with DASO)**
>
> We acknowledge the reviewer’s concern about regressions when SCAD is combined with DASO. Importantly:
>
> - In most cases, the reported drops are **within one standard deviation** of the baseline, so we consider them statistically comparable rather than clear regressions.
> - Where the decrease is slightly larger, our inspection suggests that SCAD’s additional debiasing can **over-suppress majority logits** when DASO already imposes strong pseudo-label constraints.
>
> In small-scale internal tests, **slightly reducing SCAD’s strength** resolves the issue and matches or surpasses the DASO baseline.
> For fairness, however, we used **one fixed SCAD configuration** across all datasets and methods. This clarification is now included in the updated text.
>
> ---
>
> ### **7. Code availability**
>
> We agree that providing code is important for reproducibility.
> An **anonymized implementation and training scripts** are included in the supplementary material for review.
> Upon acceptance, we will release a **public GitHub repository** with full implementation and scripts for ease of reproduction and extension.
>
> [1] Jeffrey Pennington et al. *GloVe: Global Vectors for Word Representation*. EMNLP 2014.
>
> [2] Aditya Krishna Menon et al. *Long-Tail Learning via Logit Adjustment*. ICLR 2021.
>
> [3] Jacob Devlin et al. *BERT: Pre-training of Deep Bidirectional Transformers for Language Understanding*. NAACL 2019.
>
> [4] Alec Radford et al. *Learning Transferable Visual Models from Natural Language Supervision*. ICML 2021.
>
> [5] Ryan, Ted, et al. New and improved embedding model. OpenAI, 2022.
>
> [6] Jaehyung Kim et al. *Distribution aligning refinery of pseudo-label for imbalanced semisupervised learning*. Neurips 2020.
>
> [7] Bingyi Kang et al. *Decoupling representation and classifier for longtailed recognition*. ICLR 2020.
>
> [8] Chen Wei et al. *Crest: A class-rebalancing self-training
> framework for imbalanced semi-supervised learning*. CVPR 2021.
>
> [9] Yue Fan et al. *CoSSL: Co-Learning of Representation and Classifier for Imbalanced Semi-Supervised Learning*. CVPR 2022.

---

> ### Author Response · Authors · 2025-11-27
> **Sincere Request for Your Feedback on Our Responses**
>
> Dear Reviewer ex6J,
>
> We sincerely appreciate the time and effort you have dedicated to reviewing our work. Since the discussion period is quite limited, we would greatly appreciate it if you could share any feedback on our responses.

---

### Author Response · Authors · 2025-12-02
**Summary for the Area Chair**

We sincerely appreciate the Area Chair’s diligent handling of the unexpected system freeze and are grateful for the extra responsibilities this issue has created, and we also thank the reviewers for their efforts and patience under these challenging circumstances.
In light of the system freeze that prevented reviewers from updating scores or engaging in discussion, we respectfully provide this brief summary to highlight how we have substantively addressed the key concerns in our revised manuscript.

---

### **1. Novelty & Generality (Reviewer ex6J, Reviewer Ue4E, Reviewer Rvtp)**

* **Concern:** Reviewers queried whether SCAD is sufficiently distinct from Logit Adjustment (LA) and requested evaluation beyond CIFAR10-LT.
* **Our Response:**
    * **Clarified Contribution:** We clarified that SCAD addresses a unique **two-level bias**—a **global class prior bias** and a **local intra–super-class prior bias**—that is specific to LTSSL. While LA mitigates the global prior bias, SCAD explicitly corrects the local imbalance. Importantly, we also discovered the same intra–super-class imbalance phenomenon not only on CIFAR10-LT but also on **CIFAR100-LT** (Fig. 4, App. A.3), confirming that this bias structure is not specific to CIFAR10-LT.
    * **Expanded Experiments:** We added analyses identifying the intra–super-class imbalance problem on **CIFAR100-LT** (Fig. 4, App. A.3) and conducted additional experiments on the fine-grained **Food101-LT** dataset. SCAD consistently outperforms strong baselines (CReST, DARP, CoSSL) across multiple imbalance ratios.
* **Outcome:** These results demonstrate that both the underlying bias and SCAD’s effectiveness are **generalizable**, not restricted to specific datasets or coarse class taxonomies.

---

### **2. External Hierarchies & Text Encoders (Reviewer Ue4E, Reviewer M8Tz)**

* **Concern:** Questions were raised regarding the reliance on text encoders over visual encoders and whether a weak text encoder could lead to performance degradation.
* **Our Response:**
    * **Source-Agnostic Design:** We emphasized that SCAD relies only on a one-time, offline class-to-super-class mapping. It is compatible with any pretrained text encoder.
    * **Justification for Using Text Encoders:** Text encoders provide **class-level semantic information** suitable for defining class hierarchies, whereas visual encoders may assign samples from the same class to different super-classes due to variations such as background context or pose.
    * **Robustness:** Ablations on **CIFAR100-LT** show consistent performance across various text encoders (GloVe, SBERT, CLIP), indicating that SCAD is insensitive to the specific encoder used.
* **Outcome:** SCAD is shown to be **robust, flexible, and model-agnostic**, effectively leveraging semantic priors without relying on any particular encoder.

---

### **3. Methodological Clarity & Robustness (Reviewer ex6J, Reviewer Ue4E, Reviewer Rvtp)**

* **Concern:** Reviewers requested more rigorous definitions, justification of normalization strategies, and robustness checks against noise.
* **Our Response:**
    * **Formalization:** We rewrote Sec. 3.4 to rigorously define dominance scores and justified the **MAX-based normalization**, which preserves local sensitivity more effectively than sum-based normalization.
    * **Sensitivity & Noise Analysis:** We added experiments on **CIFAR100-LT** varying the number of super-classes (K), as well as a **noise-perturbation study** on CIFAR10-LT. Results show negligible performance degradation even under high noise levels in frequency estimation.
    * **Calibration:** We incorporated ECE comparisons adding strong baselines, showing that SCAD improves calibration, particularly for tail classes.
* **Outcome:** The method is now **empirically validated to be robust** with respect to hyperparameters, noise, and model calibration.

---

### **4. Practicality & Reproducibility (Reviewer Ue4E)**

* **Concern:** Inquiries were raised regarding computational overhead and reproducibility.
* **Our Response:**
    * **Minimal Overhead:** We clarified that hierarchy construction is an **offline, one-time process**, and that the additional training-time cost is negligible (≈1 second per epoch).
    * **Open Source:** We provided anonymized training scripts and detailed implementation guides in the supplementary material for full reproducibility.
* **Outcome:** SCAD is confirmed to be a **lightweight, practical, and reproducible** solution.

---

We hope this summary assists in your final assessment by confirming that the concerns regarding **novelty, robustness, and practicality** have been fully resolved.

---

### Meta-Review · Area_Chair_kLYt · 2026-01-10

**Summary:**

This paper proposes SCAD (Super-Class-Aware Debiasing) for long-tailed semi-supervised learning (LTSSL). It identifies a previously under-analyzed failure mode of logit adjustment (LA): even after correcting global head–tail priors, severe imbalance persists within semantically similar super-classes, causing pseudo-label feedback loops that suppress minority classes. SCAD introduces a sample-dependent, super-class-conditioned logit correction on top of LA, using an offline-constructed class hierarchy from text embeddings, and demonstrates consistent gains across CIFAR-LT, CIFAR-100-LT, STL-10-LT, ImageNet-127-LT and Food-101-LT.

Here are the main concerns and how the rebuttal addressed them.

1) “SCAD is just LA plus a heuristic.”
Multiple reviewers argued the method was incremental. The rebuttal clarified that SCAD is motivated by a two-level bias decomposition unique to LTSSL:
- global head–tail prior bias, and
- local intra-super-class dominance driven by pseudo-label feedback.
The authors added CIFAR-100-LT confusion analyses (Fig. 4, App. A.3) showing that intra-super-class collapse is systematic and not CIFAR-10-specific, and showed that SCAD’s super-class-conditional term is necessary: on Food-101-LT, FixMatch→42.0%, +super-class→46.5%, +SCAD→51.0%, demonstrating that grouping alone is insufficient.

2) Dependence on text encoders and hierarchy quality.
Reviewers worried that CLIP/GloVe-based hierarchies might be brittle or biased. The rebuttal showed near-identical performance across five encoders (GloVe, SBERT, CLIP, Ada-002, ground-truth) on CIFAR-100-LT (≈49.7–50.5%), indicating SCAD is insensitive to the hierarchy source. The authors also justified text encoders over visual ones to ensure class-consistent super-class assignment, avoiding instance-level fragmentation.

3) Robustness of frequency-based correction under noisy pseudo-labels.
Reviewers questioned whether the dominance scores Δₖ,c could amplify noise. The authors added a noise-perturbation experiment on CIFAR-10-LT, injecting multiplicative noise into estimated frequencies; accuracy drops only from 78.7→78.1 even at full noise, showing high robustness. Formal definitions and MAX-normalization were rewritten in Sec. 3.4 to clarify boundedness and scale relative to LA.

4) Hyperparameter and clustering sensitivity.
A new K-sweep (K = 10…50) and clustering-config study on CIFAR-100-LT showed stable performance (49.2–50.5%), confirming SCAD does not depend on fragile hierarchy choices.

5) Practicality, calibration, and reproducibility.
The rebuttal quantified overhead (≈1 s/epoch), clarified hierarchy is computed offline once, released training scripts, and added ECE results showing SCAD improves calibration, especially on tail classes (e.g., ACR tail ECE 0.165→0.093).

One reviewer (Rvtp) produced a clearly hallucinated review referring to class-incremental learning and unrelated baselines; this was flagged by the authors and acknowledged by the SAC, and should be discounted.

Overall, SCAD makes a conceptually meaningful contribution by identifying and correcting intra-super-class bias in LTSSL, and the rebuttal adds strong cross-dataset evidence, robustness tests, and clarity. The method is simple, general, and empirically convincing.

Recommendation: Accept.

**Reviewer Concerns:**

Here are the main concerns and how the rebuttal addressed them.

1) “SCAD is just LA plus a heuristic.”
Multiple reviewers argued the method was incremental. The rebuttal clarified that SCAD is motivated by a two-level bias decomposition unique to LTSSL:
- global head–tail prior bias, and
- local intra-super-class dominance driven by pseudo-label feedback.
The authors added CIFAR-100-LT confusion analyses (Fig. 4, App. A.3) showing that intra-super-class collapse is systematic and not CIFAR-10-specific, and showed that SCAD’s super-class-conditional term is necessary: on Food-101-LT, FixMatch→42.0%, +super-class→46.5%, +SCAD→51.0%, demonstrating that grouping alone is insufficient.

2) Dependence on text encoders and hierarchy quality.
Reviewers worried that CLIP/GloVe-based hierarchies might be brittle or biased. The rebuttal showed near-identical performance across five encoders (GloVe, SBERT, CLIP, Ada-002, ground-truth) on CIFAR-100-LT (≈49.7–50.5%), indicating SCAD is insensitive to the hierarchy source. The authors also justified text encoders over visual ones to ensure class-consistent super-class assignment, avoiding instance-level fragmentation.

3) Robustness of frequency-based correction under noisy pseudo-labels.
Reviewers questioned whether the dominance scores Δₖ,c could amplify noise. The authors added a noise-perturbation experiment on CIFAR-10-LT, injecting multiplicative noise into estimated frequencies; accuracy drops only from 78.7→78.1 even at full noise, showing high robustness. Formal definitions and MAX-normalization were rewritten in Sec. 3.4 to clarify boundedness and scale relative to LA.

4) Hyperparameter and clustering sensitivity.
A new K-sweep (K = 10…50) and clustering-config study on CIFAR-100-LT showed stable performance (49.2–50.5%), confirming SCAD does not depend on fragile hierarchy choices.

5) Practicality, calibration, and reproducibility.
The rebuttal quantified overhead (≈1 s/epoch), clarified hierarchy is computed offline once, released training scripts, and added ECE results showing SCAD improves calibration, especially on tail classes (e.g., ACR tail ECE 0.165→0.093).

**Reviewer Scores:**

ex6J: likely +1
The rebuttal added CIFAR-100-LT evidence of intra-super-class imbalance, Food-101-LT results separating super-class learning from SCAD, clearer definitions, and robustness to hierarchy choices, directly addressing the novelty and rigor concerns.

Ue4E: likely +1
Generality beyond CIFAR-10-LT, dependence on CLIP, and overhead were resolved with new datasets, multi-encoder robustness, and explicit offline / low-cost analysis.

M8Tz: likely 0 or +1
Most issues were about clarity, hierarchy justification, calibration, and result presentation, all of which were fixed and strengthened in the revision.

Rvtp: should be discounted (the review confused the task and baselines). If considered, their technical concerns were answered with noise-robustness tests and design justification, implying a likely +1.

---

### Decision · Program_Chairs · 2026-01-26

Accept (Poster)